# Constructing a complete landslide inventory dataset for the 2018 Monsoon disaster in Kerala, India, for land use change analysis

Lina Hao[1,2], Rajaneesh A.[3], Cees van Westen[2], Sajinkumar K.S.[3,4], Tapas Ranjan Martha[5], Pankaj Jaiswal[6],Brian G. McAdoo[7]

[1]State Key Laboratory of Geohazard Prevention and Geoenvironment Protection, Faculty of Earth Sciences, Chengdu University of Technology, Chengdu, China
[2]Faculty of Geoinformation Science and Earth Observation (ITC), University of Twente, Enschede, the Netherlands
[3]Department of Geology, University of Kerala, Thiruvananthapuram 695581, Kerala, India
[4]Department of Geological & Mining Engineering & Sciences, Michigan Technological University, Houghton, MI, USA
[5]National Remote Sensing Centre, Indian Space Research Organisation, Hyderabad, India
[6]Geological Survey of India (GSI)
[7]Environmental Studies, Yale-NUS College, Singapore

*Correspondence to:* Lina Hao (hao_ln@qq.com) and Cees van Westen (c.j.vanwesten@utwente.nl)

**Abstract.** Event-based landslide inventories are important for analyzing the relationship between the intensity of the trigger (e.g. rainfall, earthquake) and the density of the landslides in a particular area, as a basis for the estimation of the landslide probability and the conversion of susceptibility maps into hazard maps required for risk assessment. They are also crucial for the establishment of local rainfall thresholds that are the basis of Early Warning Systems, and for evaluating which land use / land cover changes are related to landslide occurrence. The completeness and accuracy of event-based landslide inventories are crucial aspects to derive at reliable results or the above types of analysis. In this study we generated a relatively complete landslide inventory for the 2018 Monsoon landslide event in the state of Kerala, India, based on two inventories that were generated using different methods: one based on Object Based Image Analysis (OBIA) and the other on field surveys of damaging landslides. We used a collaborative mapping approach based on the visual interpretation of pre-and post-event high-resolution satellite images (HRSI) available in Google Earth, adjusted the two inventories and digitize landslides that were missed in the two inventories. The reconstructed landslide inventory database contains 4728 landslides consisting of 2477 landslides mapped by OBIA method, 973 landslides mapped by field survey, 422 landslides mapped both by OBIA and field method and an additional 856 landslides mapped using the visual image (Google Earth) interpretation. The dataset is available at https://doi.org/10.17026/dans-x6c-y7x2 (van Westen, 2020). Also, the location of the landslides was adjusted, based on the image interpretation, and the initiation points were used to evaluate the land use/land cover changes as a causal factor for the 2018 Monsoon landslides. A total of 45% of the landslides that damaged buildings occurred due to cut-slope failure while 34% of those impacting on roads were due to road cut-slope failures. The resulting landslide inventory is made available for further studies.

## 1 Introduction

Landslides are a significant type of natural hazard, occurring worldwide and incurring serious losses to human society. Landslide frequently damage buildings, communication systems, agriculture, natural vegetation, environment and are a major cause of fatalities (Froude and Petley, 2018; Petley et al., 2005). A landslide inventory forms the basis for studies of landslide hazard, risk and prevention studies (Fan et al., 2019; Guzzetti et al., 2012; Marcelino et al., 2009; Moosavi et al., 2014). Critical elements of analysis include their spatial distribution pattern (Duman et al., 2005; Galli et al., 2008; Xu, 2015), their occurrences with respect to landform evolution (Guzzetti et al., 2012; Rosi et al., 2018) and a range other environment factors (Duman et al., 2005), susceptibility mapping (van Den Eeckhaut et al., 2009), triggering factors (Li et al., 2016), community risk assessment and mitigation (Marcelino et al., 2009), and land use planning and risk management (Colombo et al., 2005). A detailed landslide inventory should contain information on location, types of failures, geometries, date of occurrence, triggering factors, possible failure mechanisms, and damage caused (Rosi et al., 2018). Landslide inventory maps can be generated by compiling existing historical landslide data or acquiring new landslide data using a variety of technical approaches (Rosi et al., 2018; Santangelo et al., 2015).

A new landslide inventory that is generated after a major triggering event, e.g. an earthquake, storm, snowmelt and volcanic eruptions, is referred to as an event-based landslide inventory (Fiorucci et al., 2011; Galli et al., 2008; Rosi et al., 2018). Methods for event-based landslide inventories include field investigation, visual interpretation of remotely sensed images, and often an automatic image classification. Field investigation shortly after the event (Fiorucci et al., 2011; Mondini et al., 2011) allows for the collection of detailed information through field surveys recording information on the location, types, volumes, contributing factors, and damages (Yang and Chen, 2010; Samodra et al., 2018). Visual interpretation based on remote sensing images (Alkevli and Ercanoglu, 2011; Mondini et al., 2011; Samodra et al., 2018), allows to map and classify landslides in terrain that is less accessible. This method will be more accurate when higher resolution images are available from the situation pre-and post the event (Li et al., 2013; Xu et al., 2013; Zhong et al., 2019). Automated classification of remote sensing images is a means to rapidly map many landslides over large areas, using different classification algorithms (Aksoy and Ercanoglu, 2012; Lei et al., 2018; Lu et al., 2019; Plank et al., 2016). Often the field surveys method are combined with remote sensing based methods to improve veracity (Ardizzone et al., 2012; Galli et al., 2008; Mondini et al., 2011; Oh and Pradhan, 2011; Rosi et al., 2018; Trigila et al., 2010).

With the continuous improvement of earth observation technology, such as multi-temporal high-resolution optical satellite remote sensing, it is more feasible to acquire remote sensing images before and after an event, which lead to more landslide inventory maps (Casagli et al., 2016; Santangelo et al., 2015; Solari et al., 2019; Travelletti et al., 2012). The availability of multi-sourced and multi-temporal high resolution satellite images (HRSI) on the Google Earth platform with 3D viewing capabilities (Crosby, 2012; Fisher et al., 2012) offered major advantages for landslide inventory mapping (Mohammadi et al., 2018). Many authors have generated landslide inventories using the Google Earth platform (Rabby and Li, 2019; Sato and Harp, 2009; Fiorucci et al., 2011; Borrelli et al., 2015). It has also proven to be possible to map event-based landslides by

65 comparing images before and after the event using Google Earth history viewer (Xu et al., 2014a, 2014b). However, recognizing and mapping specific types of landslides such as rainfall triggered shallow landslides over large areas can be still challenging when using automated techniques. Field verification is only feasible for a limited number of landslides, as it is time and labor intensive, and many landslides may be difficult to access. Therefore, visual image interpretation using HRSI from different time periods may be the best solution. Landslide mapping and classification requires mapping experience and

70 the availability of HRSI in three-dimension views, either using stereo images, or oblique views such as in Google Earth, allowing to recognize the specific diagnostic features (Soeters and van Westen, 1996; Zieher et al., 2016).

Between 1 June and 26 August 2018, the southern Indian state of Kerala witnessed the most severe extreme rainfall event since 1924 (Agarwal, 2018; Megha et al., 2019; Sankar, 2018; Vishnu et al., 2019). The torrential rains triggered several thousand landslides (Singh et al., 2018), and extensive flooding, affecting 5.4 million people in over 1,200 villages, causing

enormous property losses (buildings, roads and agriculture damages) and more than 440 casualties (Mishra et al., 2018; Vishnu et al., 2019). Furthermore, the following year, from August 8-14, 2019, Kerala was hit again by another extreme precipitation event, causing more than 100 deaths due to landslides and floods (NDTV, 2019). Due to these severe events, both the United Nations Environment Programme (UNEP) and the Government of Kerala came together to study the causes of the extreme occurrences of slope failure in this region. Concern was raised whether anthropogenic activities such as

deforestation and agriculture exacerbate the occurrence of mass movements in this region. In order to study this, a complete event-based landslide inventory is required that contains detailed land use information, to determine a causal relationship.

In this research, we generated a complete landslide inventory for the 2018 Monsoon event in Kerala, using a collaborative mapping approach based on the visual interpretation of pre-and post-event HRSI available in Google Earth, and two pre-existing inventories. This manuscript focuses on the generation of the dataset consisting of a detailed landslide inventory

with land use/land cover (LULC) information for two periods: shortly before the event, and almost a decade older. The main objective of the study is to develop a comprehensive event-based landslide inventory database for the 2018 Monsoon in Kerala that can be used to analyze to what extent these landslides were affected by land use changes.

## 2 Study area and data sources

### 2.1 Study area

Kerala is one of the most susceptible areas to mass movements in India (Sreekumar, 2009; Vasudevan and Ramanathan, 2016), with a long history for the natural occurrence of slope instability going back to 1341 AD (Kuriakose et al., 2009). Both the climate and landscape make slopes in Kerala conducive to failures. Kerala is located in the southwest of India Peninsula, in the windward slope of the Western Ghats (Sajinkumar et al., 2011), on the east coast of the Arabian Sea (Figure 1a), with typical tropical climate (the average minimum and maximum temperatures are 22 ℃ and 34 ℃) characterized by

two monsoon seasons. The southwest monsoon, lasting from June to September, delivers 80% of the annual rainfall (Paul et al., 2016), and the remainder falls in the northeast monsoon lasting from October to November. The annual average rainfall

in this area is 200 to 500 cm, which increases from the southwestern coastal plains to the mountain areas in the east due to the orographic effect of the Western Ghats (Kuriakose et al., 2009; Sajinkumar et al., 2011). Under the global climate change, extreme rainfall events have hit India frequently (Mishra et al., 2018) and the extreme rainfall events during the monsoon season are expected to increase (Hunt and Menon, 2020; Rai et al., 2019, 2020; Shashikanth et al., 2018), making it more vulnerable to slope failures.

Owing to the tropical climate, weathering of the bedrock is strong in Kerala, leading to most of this area being covered with thick poorly consolidated soil (Sajinkumar et al., 2011). The main soil is laterite with average thickness of 5 m depending on the slope (Kuriakose et al., 2009). Physiographically, Kerala can be divided into two units- a plateau with rugged mountains and deep valleys in the east and coastal plains in the west (Figure 1b) (Kuriakose et al., 2009; Sajinkumar and Anbazhagan, 2015; Vishnu et al., 2019). The Western Ghats are controlled by ancient faulted escarpments located along the plateau, often with very steep slopes, which are susceptible to slope failures (Kuriakose et al., 2009). Metamorphic rocks, such as charnockites, khondalites and gneisses are the predominant rock types in Kerala (Kuriakose et al., 2009; Sajinkumar and Anbazhagan, 2015). The combination of highly weathered bedrock and steep slopes in a monsoon climate make each district in Kerala (save the whole coastal plain district-Alappuzha) susceptible to slope instability (Figure 1) (Kuriakose et al., 2009; Sajinkumar and Anbazhagan, 2015).

## 2.2 Original data

Two landslide inventories for the 2018 triggering rainfall event were available. The first inventory came from the National Remote Sensing Center (NRSC), of the Indian Space Research Organization (ISRO) who did a rapid mapping project aimed to quickly identify slope failures in the whole Western Ghats region. They used a combination of visual image interpretation and semi-automated landslide detection based on object based image analysis (OBIA) algorithm (Martha et al., 2010, 2011, 2012, 2013, 2016). They used multi-temporal images acquired before and after the monsoon rainfall event from Resourcesat-2 and Sentinel-2 Earth observation satellites (Martha et al., 2019), resulting in an inventory with 5191 landslide polygons for Kerala (Martha et al., 2019). This rapid assessment was crucial for the emergency response by the disaster management authorities in Kerala. The fast mapping method allowed determining the general distribution, density and size of landslides in order to plan for the relief operations and overall assessment. The landslide dataset can be consulted on the Bhuvan web-platform of NRSC (https://bhuvan-app1.nrsc.gov.in/disaster/disaster.php?id=landslide_monitor). In this study, the original NRSC data was obtained as polygon shapefile (Figure 2), and then the shapefile was converted into KML for the subsequent visual interpretation in Google Earth.

Another landslide inventory was generated by the Geological Survey of India (GSI) in collaboration with the Kerala State Disaster Management Authority (KSDMA), with the aim to make a detailed survey of the landslides that specifically caused damage to buildings, roads and other infrastructure. It is important to recognize this deliberate bias in the dataset, as almost all landslides would have been mapped near roads, and is almost by definition going to be related to human occupation and transformation of the land. During a period of several months after the event, teams from GSI visited hundreds of landslide

sites. The landslides characteristics were recorded in data sheets, and transferred to spreadsheets with many attributes, including *the names of administrative units*, *latitude, longitude, types of landslide, buildings affected, road affected, recommendations, and remarks*. The GSI landslide data spanned for 10 districts (Figure 2), and the landslides studied were mainly along roads. A total of 1437 landslides points were converted into a point shapefile with all the attributes using ArcGIS 10.3, and KML for easier visual interpretation in Google Earth.

## 2.3 Problems with the use of existing inventories

After combining the above-mentioned inventories and overlaying them on multi-sourced sub-meter resolution satellite images for both the pre-and post-monsoon situation in Google Earth platform (Jacobson et al., 2015; Rabby and Li, 2019), several problems with the data were discovered through visual interpretation.

Many challenges arose when analyzing the NRSC inventory. For some of the landslide polygons no noticeable changes were visible on the images from before and after the event, and we decided to exclude these from the final inventory (Figure 3a, 3b). In some instances, there were changes visible when comparing images before and after the event which were not caused by a landslide, but by other factors, i.e. vegetation clearing (Figure 4a, 4b, 4c) or engineering activities (Figure 4d, 4e, 4f). These landslides were also excluded from the inventory. Instances were also found where the landslide polygons were displaced with respect to the landslide scarps visible in the Google Earth images (Figure 5b, Figure 6b). The NRSC landslide polygons were derived from automatic classification of Resourcesat-2 LISS IV images with 5.8 m spatial resolution, which were taken shortly after the event. Due to the coarse resolution and short time available for geocorrection, the images presented georeferencing errors, responsible for the mismatches. In these cases the landslides were mapped in the correct locations according to the Google Earth images. In other cases a polygon in the inventory was merging several smaller ones (Figure 5b), requiring us to map the individual landslides. Also the opposite was found where several smaller polygons in the inventory were part of the same large landslide (Figure 6b). Problems were also found with the use of the Google Earth images, and landslides clearly identified in the NRSC inventory could not be visually confirmed. This could be due in some areas to a long time span of 5 months between the 2018 Monsoon and the first available image after the event, which caused problems with identification due to the fast re-growth of vegetation (Figure 7b). In some cases the post-event images in Google Earth were of poor quality, due to distortion induced by steep slopes (Figure 8b, 8c), shadows induced by steep slopes (Figure 9b), or clouds obstructing the view (Figure 10b).

Also the landslide points of the GSI inventory had some problems. The surveyors marked some points where cracks or small subsidence had occurred, that did not lead to an actual landslide. These were not included in the final database. Also, some of the landslide points could not be recognized as slope failures in the images, if they were too small to be recognized or sheltered by shadows, trees or buildings (Figure 11a, 11b), or the landslide point did not match with a visible landslide scarp on the image (Figure 12b).

## 3 Methodology

### 3.1 Workflow

Since the above-mentioned problems with the available two landslide inventories would have a large influence on the analysis, we decided to correct and edit all landslides using visual interpretation based on multi-temporal HRSI available before and after the event on the Google Earth platform. These images with varying dates allow recognizing details in landforms, and land use. For those areas where the post-event images in Google Earth were distorted, obscured or missing, we used Indian Resourcesat-2 LISS-IV images (with a spatial resolution of 5.8 m and three bands of green, red and near infrared) for the earliest available post-monsoon period of 2018, which were obtained from the NRSC. By using two screens, the same landslide area was visualized using Google Earth on one screen (with KML files of the landslide points or polygons) and ArcGIS on another screen with shapefiles. With the aid of the historical image viewer tool from Google Earth, the landslides were evaluated, interpreted, assessed and measured on one screen by experts comparing multi-temporal images for the same area, while edited on the other screen for the same area. For each landslide we visually interpreted the LULC types using the Google Earth history viewer, for two time periods: before the monsoon of 2018, and for the oldest and nearly complete cover of HRSI for Kerala, which dates back to 2010. Our final landslide inventory dataset was made as points, which were carefully located on the initiation point of the landslides, with attributes related to the landslide type, and the LULC in 2010 and 2018. Due to large number of landslides in the inventory it was not possible to map the landslides as polygons, separating initiation, runout and accumulation areas (Soeters and van Westen, 1996). The workflow for the landslide inventory is shown in Figure 13.

### 3.2 Landslide mapping

The landslide mapping included the conversion of the available polygons from NRSC to points, the checking of the points from GSI) and adding new landslides that were overlooked by the available existing inventories according to the diagnostic image and geomorphological features by comparing pre-and post-event images (Borrelli et al., 2015; Fisher et al., 2012; Rabby and Li, 2019; Zieher et al., 2016). For the polygons from NRSC, the correction included the removal of erroneous polygons (Figure 3 and Figure 4), and the digitizing of a landslide point at the top of the landslide scarp (Figure 5c, Figure 6c). For areas with NRSC landslide polygons, but where post-event images in Google Earth were of poor quality or missing, a landslide point was digitized at the top of the landslide scarp based on available Resourcesat-2 LISS IV images and comparing it with the Google Earth pre-event image in 3D (Figure 7c, 9c, 10c). As the entire landslide points from GSI were mapped in the field by geologists, we only removed those points that were not classified as actual landslides but as zones with cracks and subsidence, and retained all the other points and their locations even when the landslide scarps could not be recognized on images (Figure 7a, 7b). In the locations where the GSI landslide did not match the image, we moved the landslide point to the scarp (Figure 8c). Using this procedure, the entire area was carefully checked through visual comparison of images before and after the event, and landslides that were missed in the two available inventories were added

by digitizing a point on the top of their scarp.

For each of the mapped landslide points, also a number of attributes were obtained, either from those recorded in the GSI inventory, or through visual interpretation. The following attributes were considered: landslide type, length, width, area, damage to buildings, roads and agriculture, specific reasons for failure, and the land use in 2010 and 2018.

The landslides were classified into three simple groups: shallow slide (SS), debris flows (DF) and rock fall (RF). Based on the diagnostic features described in Soeters and van Westen (1996), debris flow (DF) features were differentiated from shallow landslides (SS) by the presence of a runout zone, often reaching to the nearest stream, which is not the case for SS. Rock fall features (RF) can be differentiated from the other two processes as they occur on very steep and bare rocky slopes.

The maximum length and width were measured in Google Earth. Based on the GSI survey data and our interpretation of the satellite data we marked those landslides that caused damage to buildings, to roads, and to agricultural land. Wherever possible we identified the apparent reasons for failure through image interpretation and the atttributes from the GSI data. The following causes were identified: (1) building cutslope failure, (2) road cutslope failure, (3) inadequate drainage along the road, (4) reactivation of old landslides, (5) undercutting of slope by river, (6) reservoir increase causing instability along the slopes, (7) deforestation, (8) clearing of tea plantation, (9) clearing of rubber plantation, (10) the margin area between different land use types.

### 3.3 Land use attributes

To study the relation between landslides and recent land use changes, detailed and precise land use information immediately before the 2018 event was required, together with land use information for some time earlier. The available online land cover products, such as IGBP DISCover, UMD Land Cover, Global Land Cover 2000 and GlobCover 2009 (Congalton et al., 2014), have too coarse resolution for a proper correlation with the landslides (Seo et al., 2014). Several historical digital land use maps from Kerala were also available from the KSDMA, however, after careful comparison with the corresponding HRSI using Google Earth history viewer, we decided not to use them because of the insufficient spatial and thematic accuracy. Figure 14a illustrates this by overlaying the 2010 land use map on the HRSI of the same year. The first problem is that the land use polygons do not match the image information (i.e. the shape of polygon A and B do not match with the image from the same year). The second problem is that the land use polygons in this 1:50.000 scale land use map are too generalized for analyzing specific relations with landslides. One land use polygon may contain more than one land use type, i.e. the land use type of polygon B is *Tea*, while on the detailed images it can been interpreted that it contains roads, buildings, shrubs, bare farm land, and forest as well (Figure 14a). If this map was used for correlating landslide occurrences with land use types, the land use type in 2010 of all landslides in Figure 14b would have been *Tea*. However, the actual land use types were bare farmland (landslide I, III and IV) and shrub plantation (landslide II and V) (Figure 14c).

In order to correlate landslide occurrences with the land use (change) at specific locations (like landslide scarps), detailed and accurate land use data are needed. Automatic image classification would not give the required accuracy and detail (Srivastava et al., 2012), due to the complexity of the terrain and the detailed land use legend needed. It has proven very

difficult to differentiate natural land use types (e.g. forest) from cultivated area (e.g. mixed forest plantations) using automatic image classification. Automatic image classification also requires a large number of very high-resolution cloud free images for at least two periods covering the whole landslide affected area of Kerala, which require costs that were beyond the scope of this project.

In view of the above problems, we decided to visually interpret the land use types for each landslide initiated area based on the Google Earth history viewer, in which the oldest and nearly complete cover of HRSI for Kerala dates back to 2010. Visual interpretation is useful in land use mapping (Butt et al., 2015; Mohammady et al., 2015; Kibret et al., 2016) with higher accuracy (Audah et al., 2019; Ghorbani and Pakravan, 2012), especially in complicated areas (Huang et al., 2018). A skilled interpreter, who is familiar with land use types and was trained to identify diagnostic features of various land use

types in the study area, is able to extract detailed land use information from the image interpretation elements of pattern, shape, context, size, shadows, phenology, spatial relation, and changes (Cihlar and Jansen, 2001), as well as using clues from available land use maps from NRSC for differentiating cultivation from natural vegetation. Differentiating agriculture from natural vegetation was considered important to model relationship between landslides and land use. An interpreter will generally be able to discriminate the boundaries of complicated land use types with a higher accuracy than can be obtained

through automatic classification, although it will take much more time (Miettinen et al., 2019).

The use of Google Earth history viewer allows to frequently compare the temporal image characteristics of the same area using vertical as well as oblique views in different directions, which are all helpful in recognizing land use types. Furthermore, the land use in the direct surrounding of the landslide can be interpreted as well, allowing the interpreters to make a better evaluation of the relation between land use and landslides. For each landslide the land use situation was

evaluated for the year 2010 and for the year 2018, prior to the occurrence of the extreme event in August 2018. Mapping was done as a collaborative exercise, involving a group of four mappers. A detailed legend was worked out first and discussed among the mappers, in order to achieve a standard interpretation. Also, regular cross-checks were made of each other's results to ensure a standardized approach. The ability to visually differentiate land use types was taken into account in defining the land use legends (Fox et al., 2017). Land use /land cover types were selected in such a way that they differed

with respect to their influence on landslides, in terms of vegetation cover, anthropogenic activities, hydrological effects and the characteristic vegetations' roots (Karsli et al., 2009; Reichenbach et al., 2014). Ultimately, twenty five land use types were defined in our study (see Figure 16). For each landslide point on the top of a landslide scarp, the historical image viewer of Google Earth was used to visualize the surrounding areas before failure in 2018 using the earliest available images, and the land use situation around 2010 (using the image that is closest to this period).

**4 Resulting landslide inventory**

**4.1 Complete landslide inventory for the 2018 Kerala Monsoon event**

After the landslide mapping and attribute editing, a complete landslide point inventory dataset for the 2018 Monsoon event

in Kerala was generated, containing 4728 confirmed landslides. Out of these, 2477 landslides (52%) were derived from the NRSC polygons, and 973 landslides (21%) from the GSI points with, 422 landslides (9%) that were included in both inventories. Additionally, 856 new landslide points (18%) were identified using HRSI available in Google Earth (Table 1, Figure 15a).

The most common landslide type was *debris flow* (DF: 2816 landslides), followed by *shallow slide* (SS: 1760) and *rock fall* (RF: 152) (Table 2, Figure 15b). The landslide types for the NRSC inventory were interpreted by us using the visual mapping of the Google Earth images. They differed from the GSI landslide inventory, with a higher proportion of debris flows in the NRSC data (71% of the polygons were DF, 25% SS and 4% RF) as compared to the GSI mapped slides (44% DF, 55% SS and only 1% RF). The Idukki district was mostly affected by landslides, accounting for 47.02% of the total landslides in Kerala (Figure 15).

Figure 16 shows the frequency of landslides for the different land use/land cover types in 2010 and 2018. The results show that the highest proportion of the landslides were initiated in Mixed Plantation Forests (FMP, 25.06%), followed by Dense Natural Forests (FDN, 23.33%). This is an interesting result in view of the expectation that forests are less vulnerable to landslides, due to the hydrological and geomechanical characteristics of trees which tend to reduce the chance of slope stability (Alcántara-Ayala et al., 2006; Reichenbach et al., 2014; Tasser et al., 2003). Also a significant percentage of 14% of all landslides occurred in steep areas with bare rock and soil and sparse vegetation.

Among all the landslides in this event in Kerala, 2503 out of 4728 landslides caused damages to buildings, roads and agricultures, accounting for 52.94%. Apart from the 1205 damaging landslides surveyed in the field by GSI, the image interpretation revealed another 90 landslides with damage to buildings, 356 with damage to roads, and 1251 with damage to agriculture (Figure 17). As for building impacts, 645 landslides destroyed 942 buildings, of which most were residential buildings (Figure 17). A shallow slide (SS) in Kannur damaged 23 buildings while a debris flow (DF) in Wayanad destroyed 12. Landslides associated with building cut-slopes were responsible for 45% of the damaged buildings. Regarding road impacts, 897 landslides caused traffic disruption after the event, among which 625 landslides covered roads which need to be cleared while 272 landslides damaged roads that had to be repaired. Landslides associated with road cut-slopes were responsible for 34% of the road impacts. For agriculture impacts, 2194 landslides damaged the agricultural land use classes of TEA, FMP, RUB, SPL and FCP (Figure 16). FMP, SPL and TEA suffered the most damages of all cultivation land.

The results show that only a relatively small number of landslides (707, 14.95%) were located in sites where land use changes occurred in the past eight years before their occurrence (Figure 16). The vast majority of the landslides were not related to land use changes in the past decade.

### 4.2 Comparison of inventories

The final landslide dataset was made by integrating two inventories that were acquired using different methods. In the final inventory, 2899 (61.32%) out of 4728 landslides were obtained directly from the results of the automatic classification, which were accepted after careful visual interpretation of multi-temporal HRSI. Among the 2899 landslides, 2657 landslides

were mapped as points directly from an equal number of polygons, 163 landslide points were made by merging 366 polygons (when several polygons belonged to the same landslide), and 79 landslides were mapped by separating 35 polygons (when a single polygon contained several landslides). Only 422 out of 1437 landslides with confirmed damage, mapped by GSI, were identified by automatic image classification.

In the final landslide point dataset, 1276 (27%) out of 4728 landslides were confirmed only by one source, while a total of 3452 (73%) landslides were confirmed by at least two independent sources (Table 3). Among the single sourced 1276 landslides, 420 (9%) landslides without an estimation of the area of the landslides, as those were the points from GSI for which no area could be determined in the images, because the landslides were too small. These 420 landslides were mapped by GSI as they caused damage to buildings and roads, but could not be identified on Google Earth or Resourcesat-2 satellite images, due to the small size or sheltering by buildings, trees, and clouds. Still, they are accepted in the final dataset because they were visited by geologists in the field. The rest of 856 (18%) single sourced landslides were identified and confirmed by their clear signs on multi-temporal Google Earth images, and about 25 of these were confirmed by field investigation by the authors in May, 2019. Therefore the minimum overall accuracy of the final inventory is 73%, although we consider it to be much larger, given the fact that we visually inspected the entire area. However, it is not possible to quantify the completeness of the final inventory, due to the lack of another independent and confirmed complete inventory.

## 5 Data availability

The landslide dataset, and a document with metadata, is freely downloadable from https://doi.org/10.17026/dans-x6c-y7x2 (van Westen, 2020) and available for further analysis. The landslide dataset is provided in the form of an ESRI point shapefile including the following attributes: *district*, *landslide type*, *area*, *damage* (building impact, road impact, and agriculture impact), *land use in 2010*, *land use in 2018*, *specific reasons for landslide occurrence*, *remarks* and *data source*. The definition of each attribute and the codes are provided in an accompanying metadata Word document. The dataset aims to contribute to further understanding of the relation between rainfall intensities and associated spatial distribution of landslides, in order to improve the methods for rainfall-induced landslide hazard assessment, and the development of more accurate rainfall thresholds for early warning. The dataset also aims to contribute to further research on the relation between land use changes and landslide occurrences, which is also an important aspect, especially due to the observed increase in extreme hydro-meteorological hazard events.

## 6 Discussion and conclusions

The results show that more than half of the damaging landslides (613) surveyed by GSI, were very small (<500 m$^2$). Many of these small-sized landslides could not be visually identified and measured even on HRSI, as they may be covered by dense vegetation or sheltered by buildings and other objects. This makes it also very difficult to detect them using automatic image

classification, as no more than half of these damaging landslides (422 out of 973) were detected. This is an important factor as the automatic image classification provides a rapid survey of the possible landslide area, soon after the event. Reconnaissance in the field by geologists is the best method for mapping such landslides (Brardinoni et al., 2003). The survey requires considerably more time and resources, and it took survey teams of 20 persons one month to carry out the survey, with a follow up survey by 10 persons for another three months. The survey was also biased towards damage along the roads. Although time consuming and biased toward to landslides close to road, field-based surveys remain an essential component for the damage assessment and post-disaster recovery as it will obtain quantitative information on the damage caused by landslides and will neither be replaced by image interpretation nor automatic image classification (Moosavi et al., 2014).

For landslides with an area larger than 1000 m$^2$, automatic image classification is a very useful tool, as evidenced by this study, where more than 76.3% of all large landslides were detected automatically. The automatic classification method is useful for detecting landslide with a certain minimum size (Lahousse et al., 2011; Martha et al., 2011) depending on the resolution of remote sensing images (Fiorucci et al., 2011; Harp et al., 2011). OBIA is very effective for generating a rapid first inventory of larger landslides triggered by an event such as intense rainfall and earthquake (Behling et al., 2014; Lu et al., 2011; Martha et al., 2016). However, the accuracy of these automatic recognition methods still need to be improved (Feizizadeh et al., 2017), and care should be taken to derive statistical relationships with causal factors from such inventories due to the significant overestimation of the number of landslides, and because the relations would only be meaningful for the initiation areas of the landslides, and not for the full polygon areas that are normally identified using OBIA.

During this Monsoon event triggered landslide inventory, it took teams of 6 persons 39 days (one person works 8 hours per day) for the visual interpretation checking and digitizing. Comparing automatic image classification, visual interpretation of satellite data is a cost-effective, yet quite time consuming method for mapping event-triggered landslides (Yu and Chen, 2017), and has a high accuracy if combined with field investigation (Fiorucci et al., 2011; Mondini et al., 2011). Also, landslides above a minimum size of 20 m$^2$ can be recognized based on sub-meter HRSI, if they are not masked by shadows of nearby slopes, objects or vegetation. The comparison of pre-and post-event satellite images, and the integration with the results of automatic image classification in a platform such as Google Earth history viewer, was very useful for the generation of a complete and reliable inventory. The collaborative mapping approach, involving a number of mappers, in different locations, required a good communication and cross-checking of the interpretation results, to ensure consistent results among the mapper, but reduced the mapping time comparing to field-based survey method, and the costs for image acquisition were greatly reduced by using Google Earth images (van Westen et al., 2008).

The Monsoon event of 2018 in Kerala killed more than 483 persons (Sahana, 2019), triggered 4728 landslides, which damaged 942 buildings. It was an extreme event, and the damage was attributed in popular literature to climate change and anthropogenic changes, especially the decrease of natural forests and the increase of buildings in sloping terrain (The Conversation, 2019). Recent studies (Ramachandra and Bharath, 2019) have analysed that the forest cover in the Western Ghats has decreased by 30%, from 16.21% in 1985, to 11.3% in 2018. The region now has 17.92% plantation area, 37.53%

agriculture and 4.88 % mining and built-up urban areas. It is therefore remarkable that the majority of the landslides triggered during the 2018 monsoon event occurred within forested areas. Also for the vast majority of the landslides no significant changes in land use were detected in the past 8 years, suggesting that this was indeed an extraordinary rainfall event where land use played a relatively minor role. Further research is needed to study the intricate relations between land use change and landslide occurrence.

So far, the final inventory of significant landslides those that damaged buildings, roads, or agricultural areas as well as failures large enough to be seen in various satellite images, can be considered relatively complete for the 2018 event, as the entire area was carefully checked using multi-temporal visual image interpretation. However, it is possible that a few landslides were still missed in the final dataset due to the very small size and shelter. It is not possible to quantify the completeness of the final inventory, due to the lack of another independent and confirmed complete inventory.

**Author contributions**

LH, CW designed the work together with SKS. LH and RA compiled the dataset. LH wrote the manuscript supervised by CW. TRM and PJ provided some of the data, and suggestions on some methods. LH, RA and SKS performed the figures. CW and BGM provided suggestions on structure, methods and figures. All the authors contributed to the review of the manuscript and approved the dataset.

**Competing interests**

The authors declare that they have no conflicts of interest.

**Acknowledgements**

We thank the United Nations Environmental Programme (UNEP) (Muralee Thummarukudy, Karen Sudmeier-Rieux, and Louise Schreyers) for initiating this work and the coordination; Sekhar Lukose Kuriakose and colleagues from the Kerala State Disaster Management Agency (KSDMA) for their support; the National Remote Sensing Centre (NRSC) for providing the landslide polygon inventory; the Geological Survey of India (GSI) for providing the landslide point inventory; and Google Earth for the use of multi-temporal HRSI. The research was also co-funded by UNEP and the Chinese National Science Fund (grant no. 41702358, 41790445, 41630640, 41771444) and the China Postdoctoral Science Foundation (grant no. 2017M622982).

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

**Table 1: The number of landslides per district in Kerala of the final dataset with various sources**

| District        Source | NRSC | GSI | GSI+NRSC | New | Total /% |
|---|---|---|---|---|---|
| Idukki | 607 | 685 | 256 | 675 | 2223/47.02 |
| Pathanamthitta | 66 | 24 | 7 | 9 | 106/2.24 |
| Kottayam | 43 | 18 | 13 | 2 | 76/1.61 |
| Thrissur | 206 | 33 | 17 |  | 256/5.41 |
| Ernakulam | 94 | 10 | 3 |  | 107/2.26 |
| Palakkad | 649 | 54 | 36 | 54 | 793/16.77 |
| Kozhikode | 97 | 23 | 18 | 90 | 228/4.82 |
| Malappuram | 312 | 59 | 36 | 22 | 429/9.07 |
| Wayanad | 250 | 53 | 26 | 2 | 331/7.00 |
| Kannur | 116 | 14 | 10 | 1 | 141/3.00 |
| Kasaragod | 24 |  |  |  | 24/0.51 |
| Kollam | 10 |  |  |  | 10/0.21 |
| Thiruvananthapuram | 3 |  |  | 1 | 4/0.09 |
| Total | 2477 | 973 | 422 | 856 | 4728 |

**Table 2: The number of landslide classified by types in each district in Kerala**

| District        Type | Shallow Slide (SS) | Debris Flow (DF) | Rockfall (RF) |
|---|---|---|---|
| Idukki | 1421 | 679 | 123 |
| Pathanamthitta | 13 | 92 | 1 |
| Kottayam | 11 | 65 |  |
| Thrissur | 20 | 234 | 2 |
| Ernakulam | 11 | 96 |  |
| Palakkad | 92 | 699 | 2 |
| Kozhikode | 18 | 204 | 6 |
| Malappuram | 66 | 358 | 5 |
| Wayanad | 68 | 252 | 11 |
| Kannur | 19 | 120 | 2 |

| | | |
|---|---|---|
| Kasaragod | 19 | 5 |
| Kollam | 1 | 9 |
| Thiruvananthapuram | 1 | 3 |
| Total | 1760 | 2816 | 152 |

**Table 3: The number of landslide confirmation by different means in Kerala**

| Confirmation means | GSI only (Field mapping) | Google Earth only (Visual image interpretation) | GSI, Google Earth | NRSC, Google Earth/ Resourcesat-2 LISS-IV | GSI, NRSC, Google Earth |
|---|---|---|---|---|---|
| Number/% | 420/9% | 856/18% | 553/12% | 2477/52% | 422/9% |

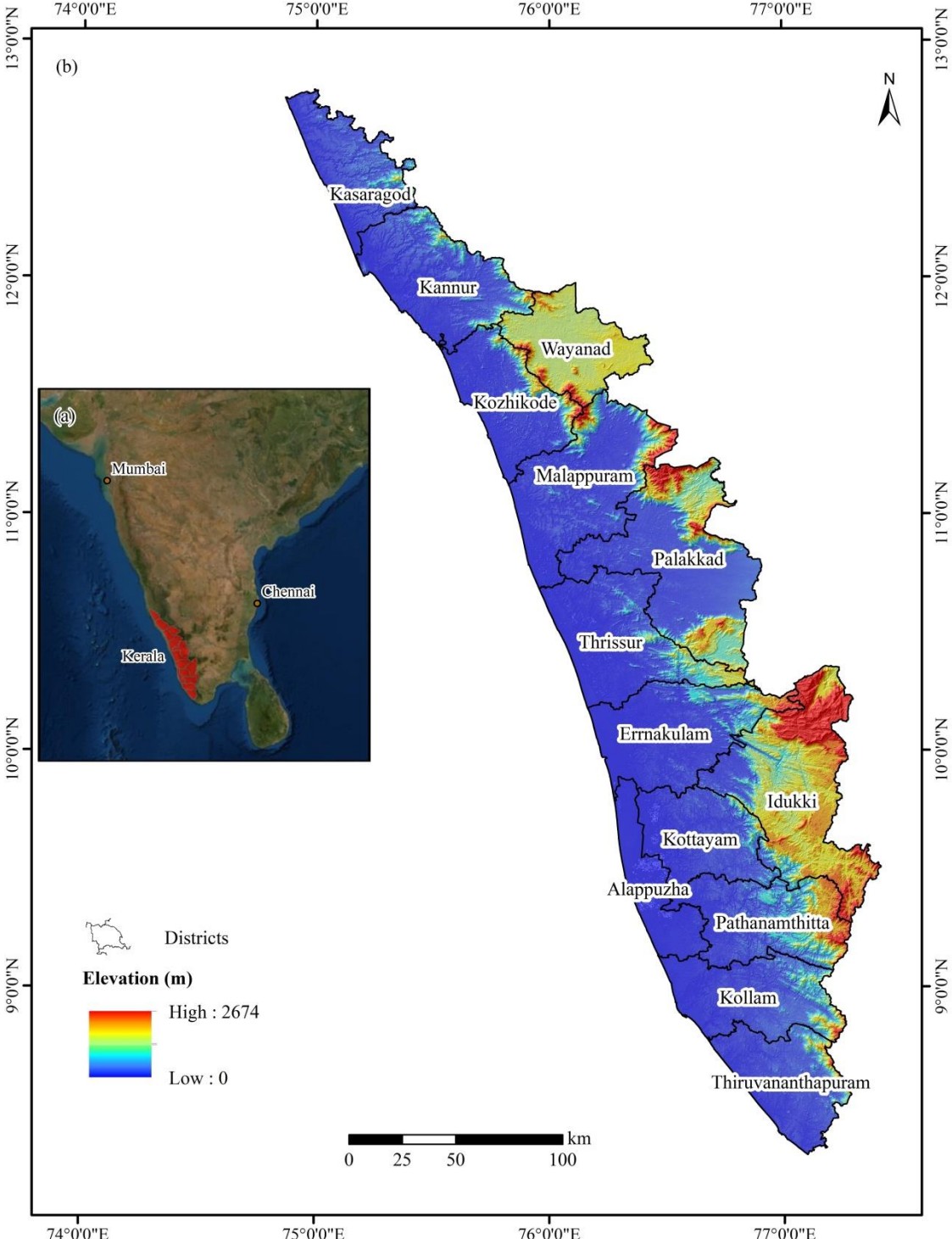

**Figure 1: Overview map of Kerala with districts and elevation.**

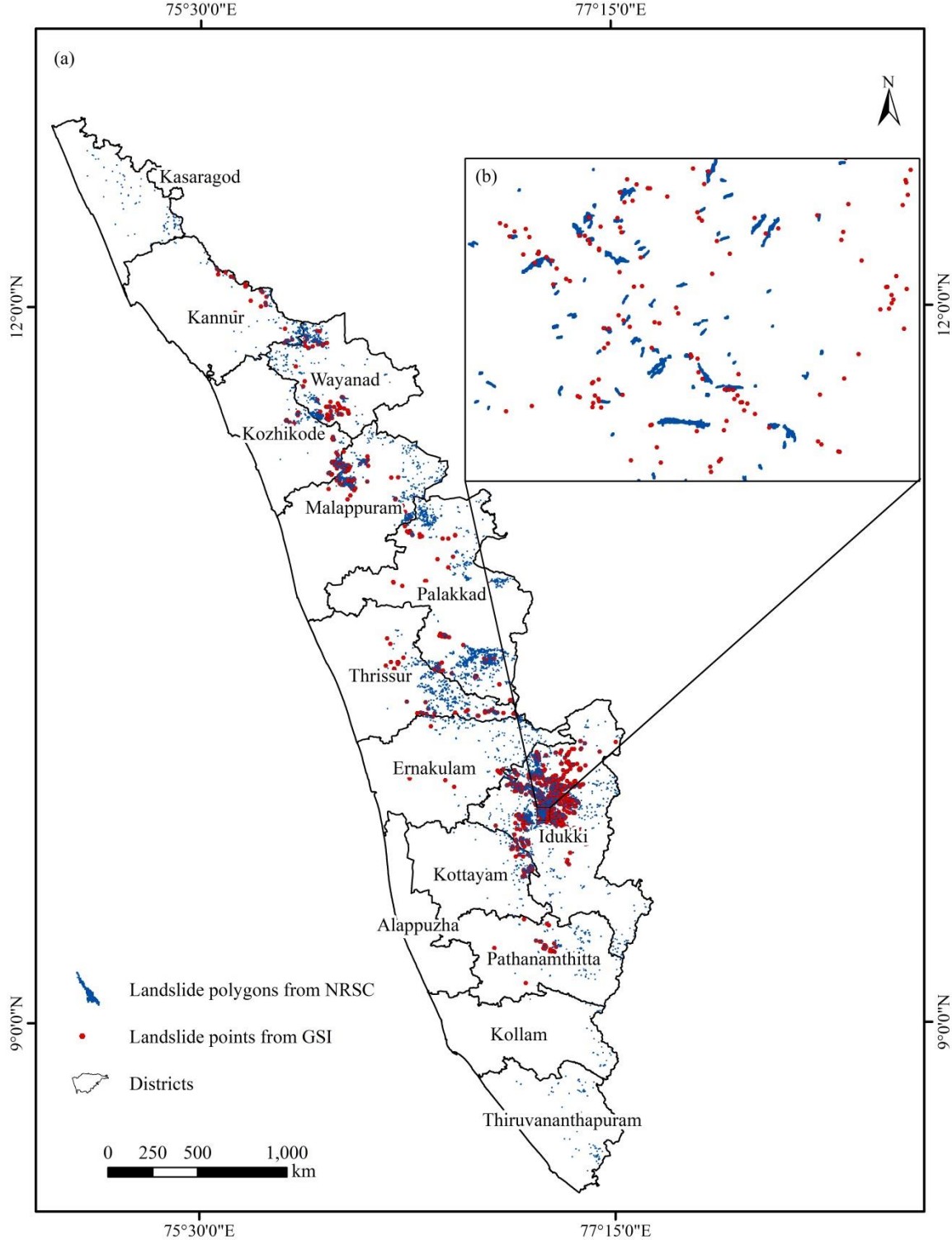

**Figure 2: Overview map of the existing inventories.**

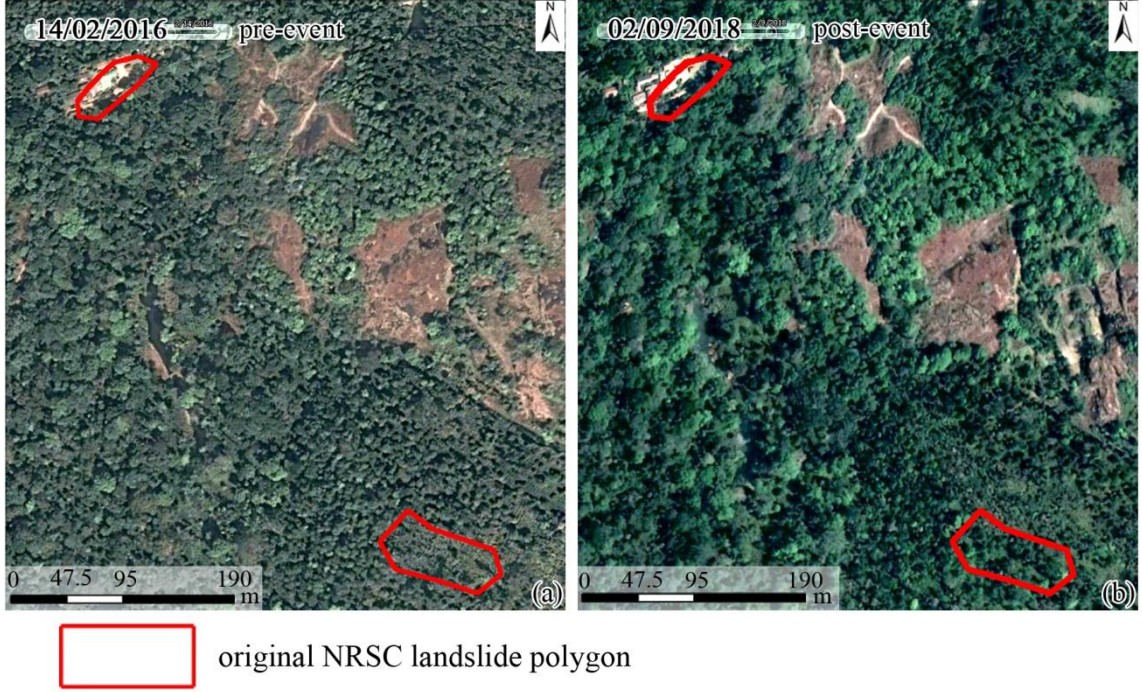

original NRSC landslide polygon

**Figure 3: Examples of landslides in the NRSC inventory that were not considered as actual landslides after visual inspection. The examples in (a) and (b) show that there are no visible scarps before and after the event near the marked polygons. Basemap data© 2019 Google**

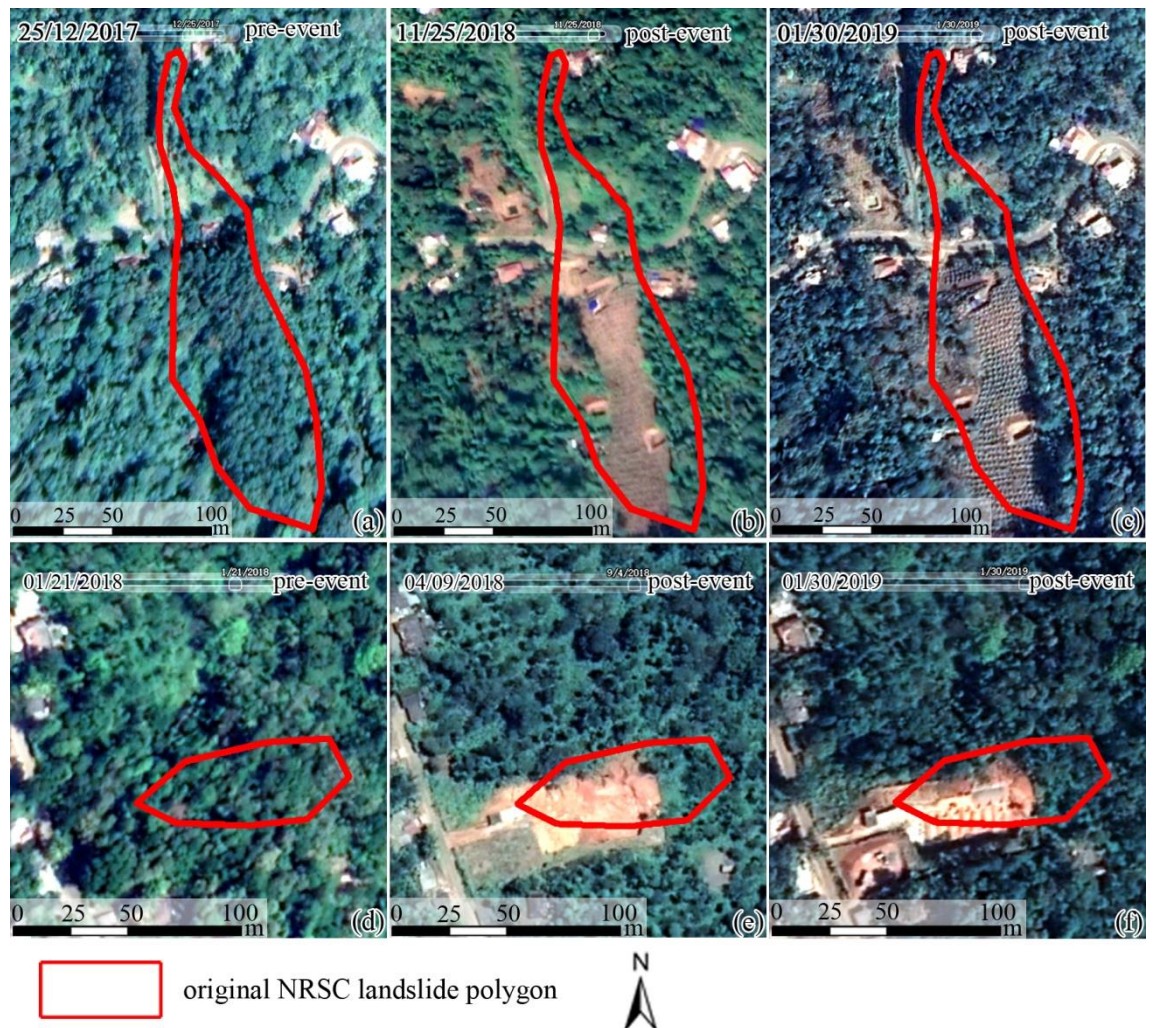

original NRSC landslide polygon

**Figure 4: Examples of landslides in the NRSC inventory that were not considered as actual landslides after visual inspection. The example in (a), (b), and (c) shows that the changes in the polygon before and after the event were caused by vegetation clearing, and agricultural activities. Images (d), (e), and (f) show that the changes near the polygon before and after the event were caused by building construction. Basemap data© 2019 Google**

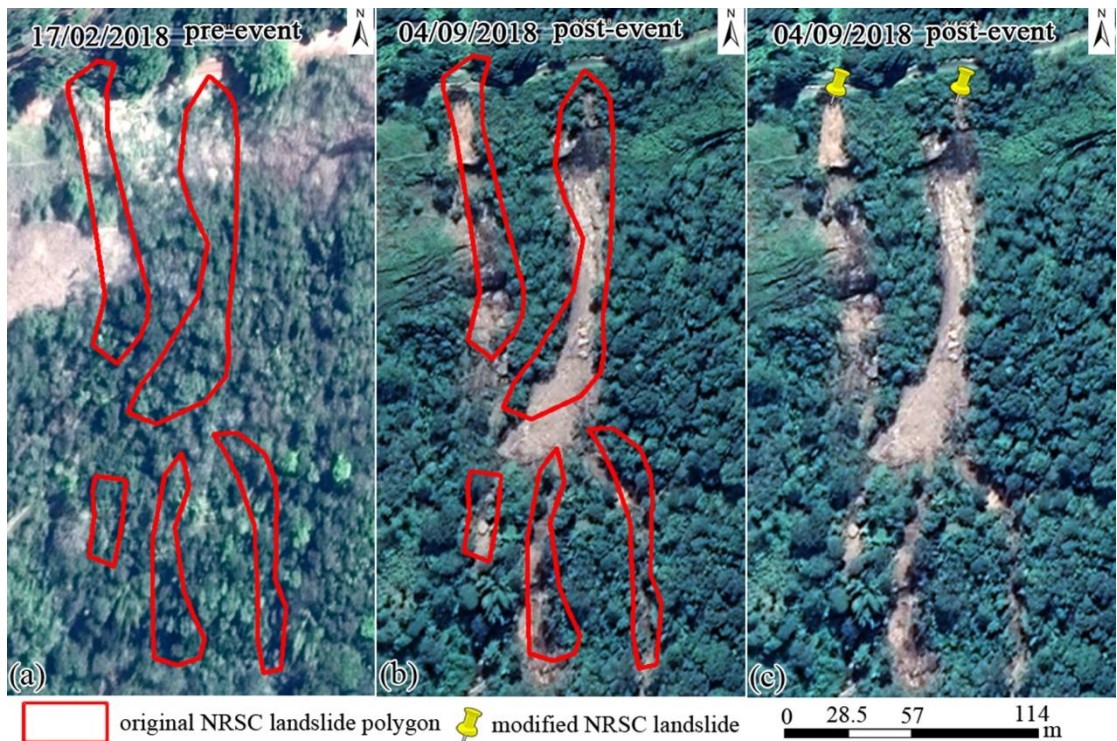

**Figure 5: Example of a situation where the original NRSC landslide polygons were combined and converted into points and digitized on the top of the scarps. (a) pre-landslide image; (b) post-landslide image; (c) final inventory using points. Basemap data© 2019 Google**

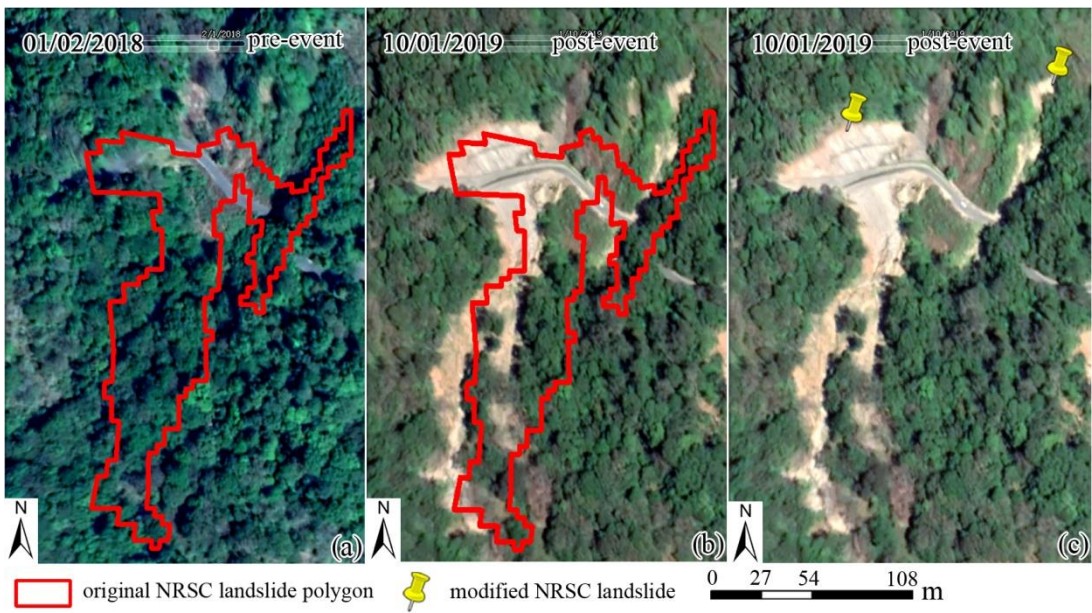

**Figure 6: Example of a situation where the original NRSC landslide polygon was subdivided into several landslides, marked by points and digitized on the top of the scarps. (a) pre-landslide image with NRSC landslide polygon on top; (b) post-landslide image,**

**with the NRSC polygon on top, which shows that there are two landslides instead of a single one; (c) the mapping of landslide points in the scarps of the two landslides. Basemap data© 2019 Google**

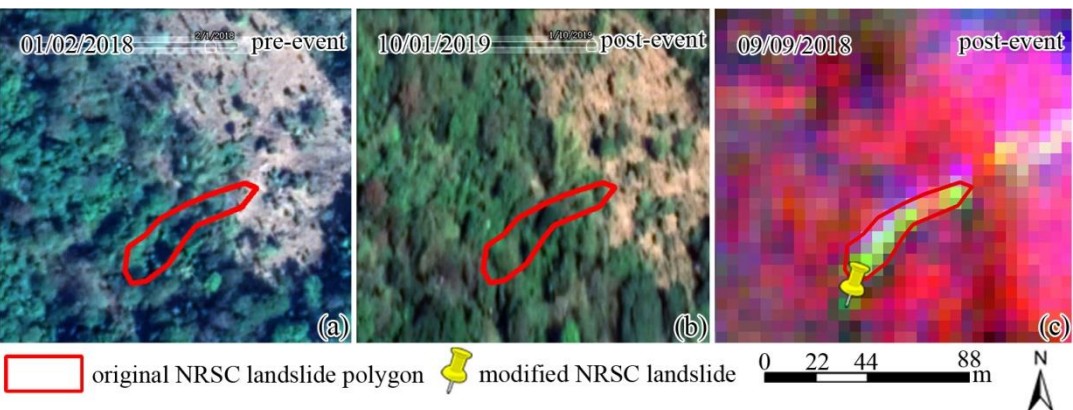

Figure 7: Example of a situation where vegetation re-growth made it difficult to identify the scarps on Google Earth images due to the large time span between the event and the first available images within Google Earth. The original NRSC landslide polygons were generated from the classification of Resourcesat-2 LISS-IV images that were taken within 15 days of the event. (a) pre-landslide Google Earth image; (b) earliest available post-landslide Google Earth image, where the landslide cannot be recognized; (c) mapping of the landslide initiation point based on Resourcesat-2 LISS-IV image (RGB combination: near infrared, red, green). Basemap data for (a) and (b) © 2019 Google

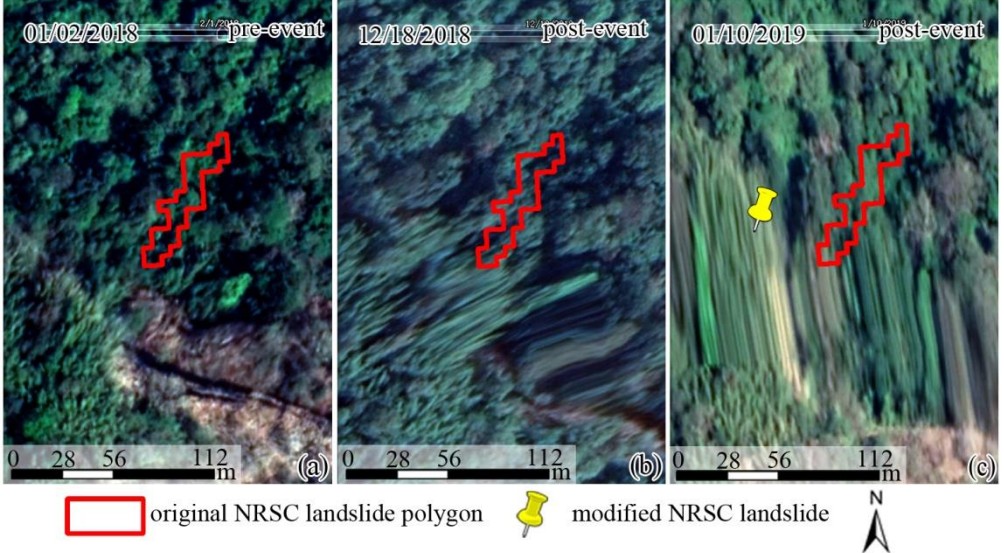

**Figure 8: Example of distorted images in Google Earth where it was not possible to check the original NRSC landslide polygons. They were converted into points and digitized on the top of the scarps based on Resourcesat-2 LISS-IV images. (a) pre-landslide image; (b) post-landslide image; (c) creation of a new inventory using points based on post-landslide. Basemap data© 2019 Google**

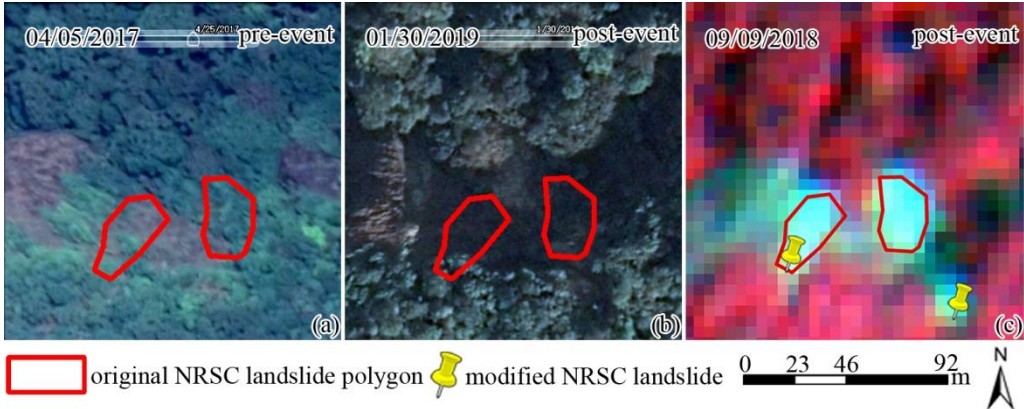

**Figure 9: Example of the presence of darks shadows in the post-event images in Google Earth images, making it impossible to check the original NRSC landslide polygons. (a) pre-landslide Google Earth image; (b) earliest available post-landslide Google Earth image, where the landslide cannot be recognized; (c) mapping of the landslide initiation point based on Resourcesat-2 LISS-IV image (RGB combination: near infrared, red, green). Basemap data for (a) and (b) © 2019 Google**

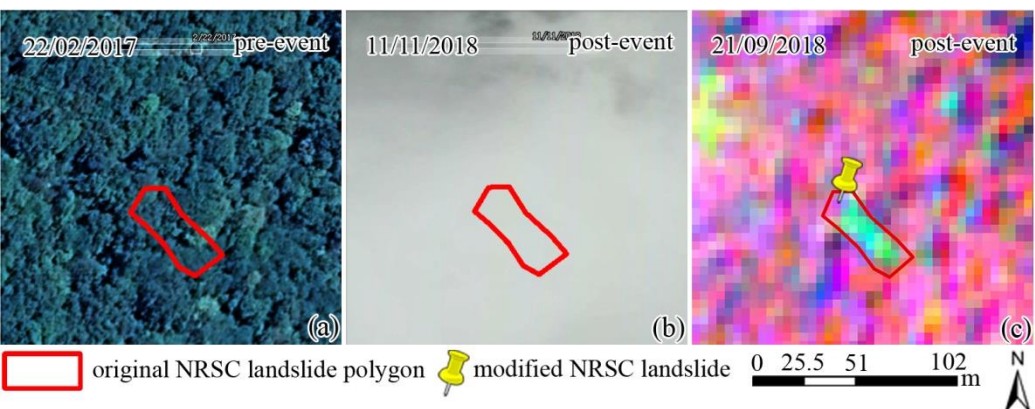

**Figure 10: Example of the obstruction of view by clouds where the original NRSC landslide polygons could not be checked. (a) pre-landslide Google Earth image; (b) earliest available post-landslide Google Earth image, where the landslide cannot be recognized; (c) mapping of the landslide initiation point based on Resourcesat-2 LISS-IV image (RGB combination: near infrared, red, green). Basemap data for (a) and (b) © 2019 Google**

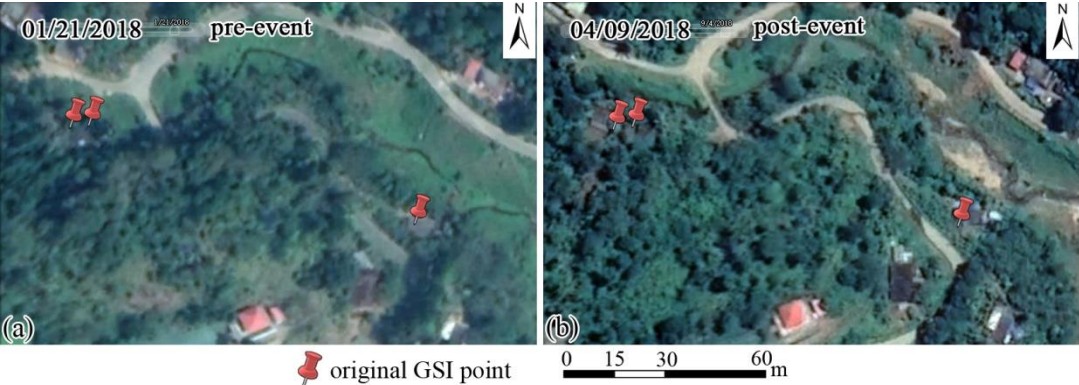

**Figure 11:** Example of a situation where the original GSI landslide points were accepted although there were no manifestation of landslide scarps was visible in pre- and post- event images within Google Earth. We assumed that landslide were properly marked in the field by the surveyors, and that they must have been very small and hidden from view by surrounding vegetation or buildings. Basemap data© 2019 Google

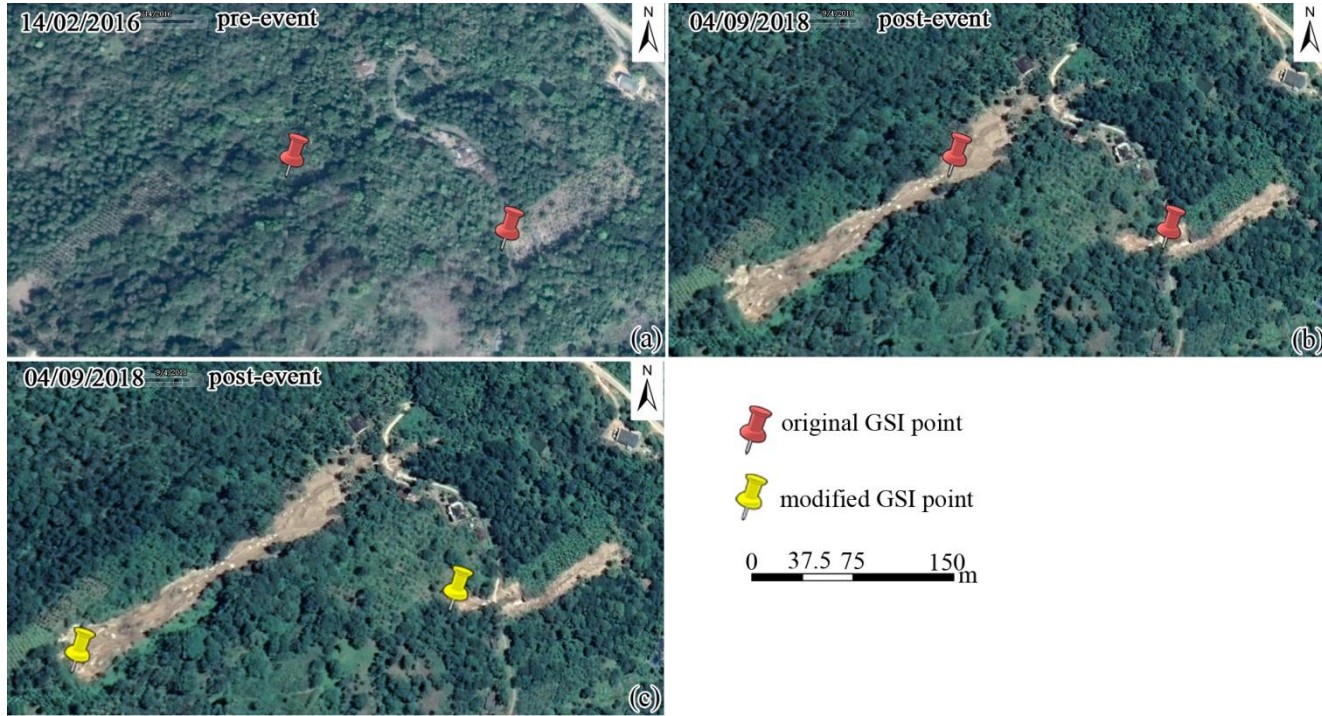

**Figure 12:** Example of a situation where the original GSI landslide points were shifted to the top of the landslide scarps. (a) pre-event image with landslide points from the GSI inventory; (b) post-event image with original landslide points from the GSI inventory; (c) post-event image with adjusted landslide points. Basemap data© 2019 Google

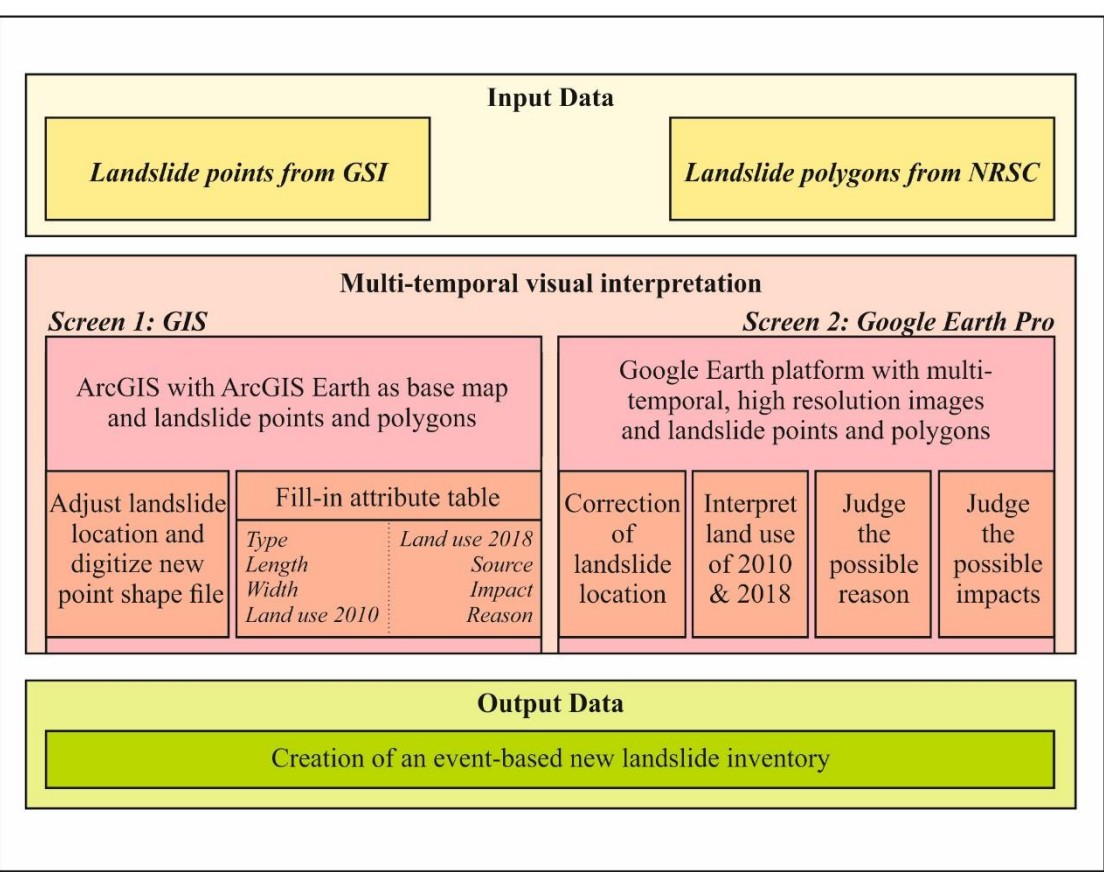

**Figure 13: Overview of the methodology adopted for the creation of a new landslide inventory in this study.**

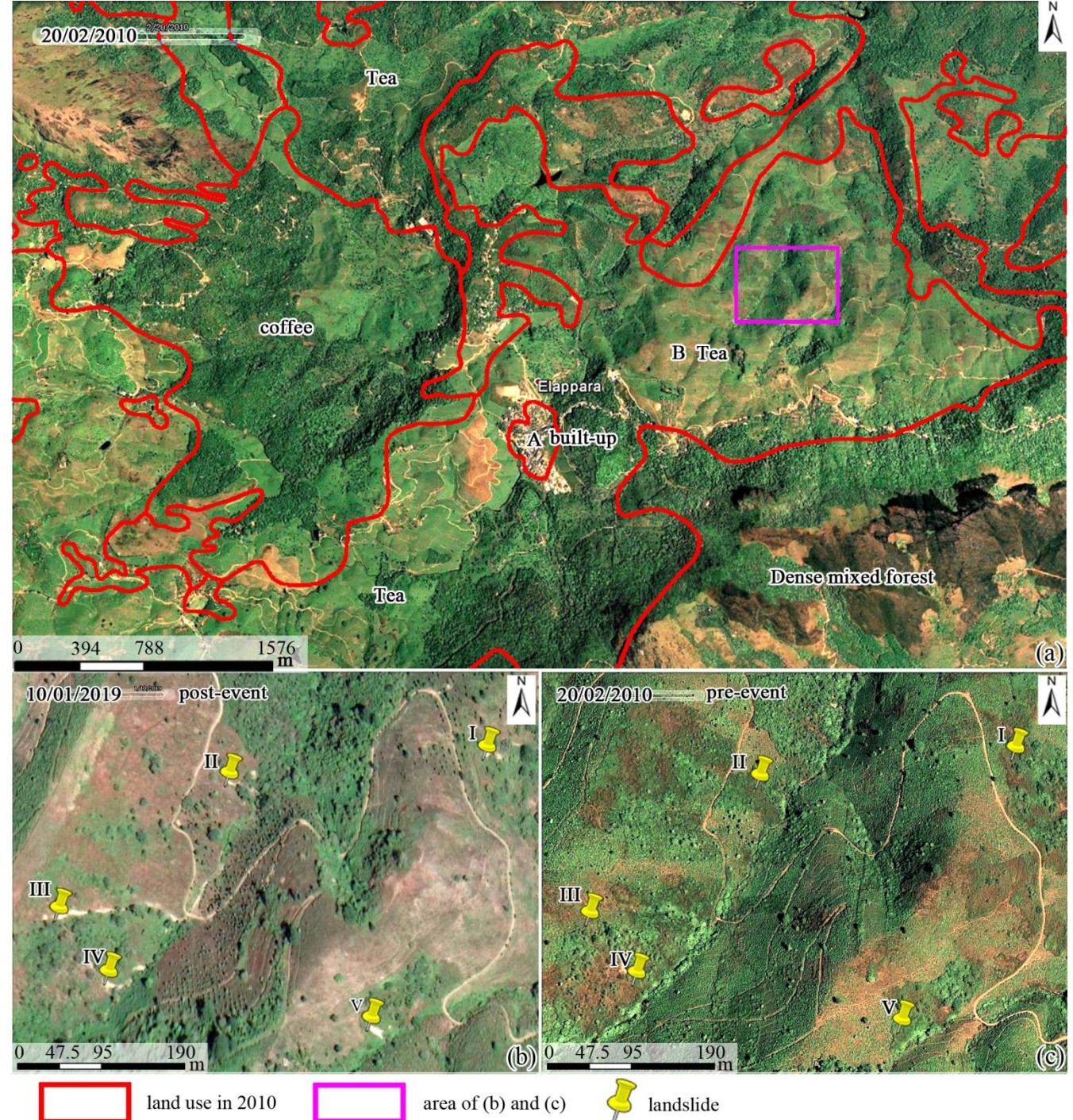

**Figure 14: Example of the problem in using the available land use map. The boundaries of the available 1:50,000 land use map from 2010 is shown on a HRSI of the same year. The detailed images shown in (b) and (c) contains many more land use types, than the single one indicated in the map, leading to wrong correlations between landslides and land use. Basemap data© 2019 Google**

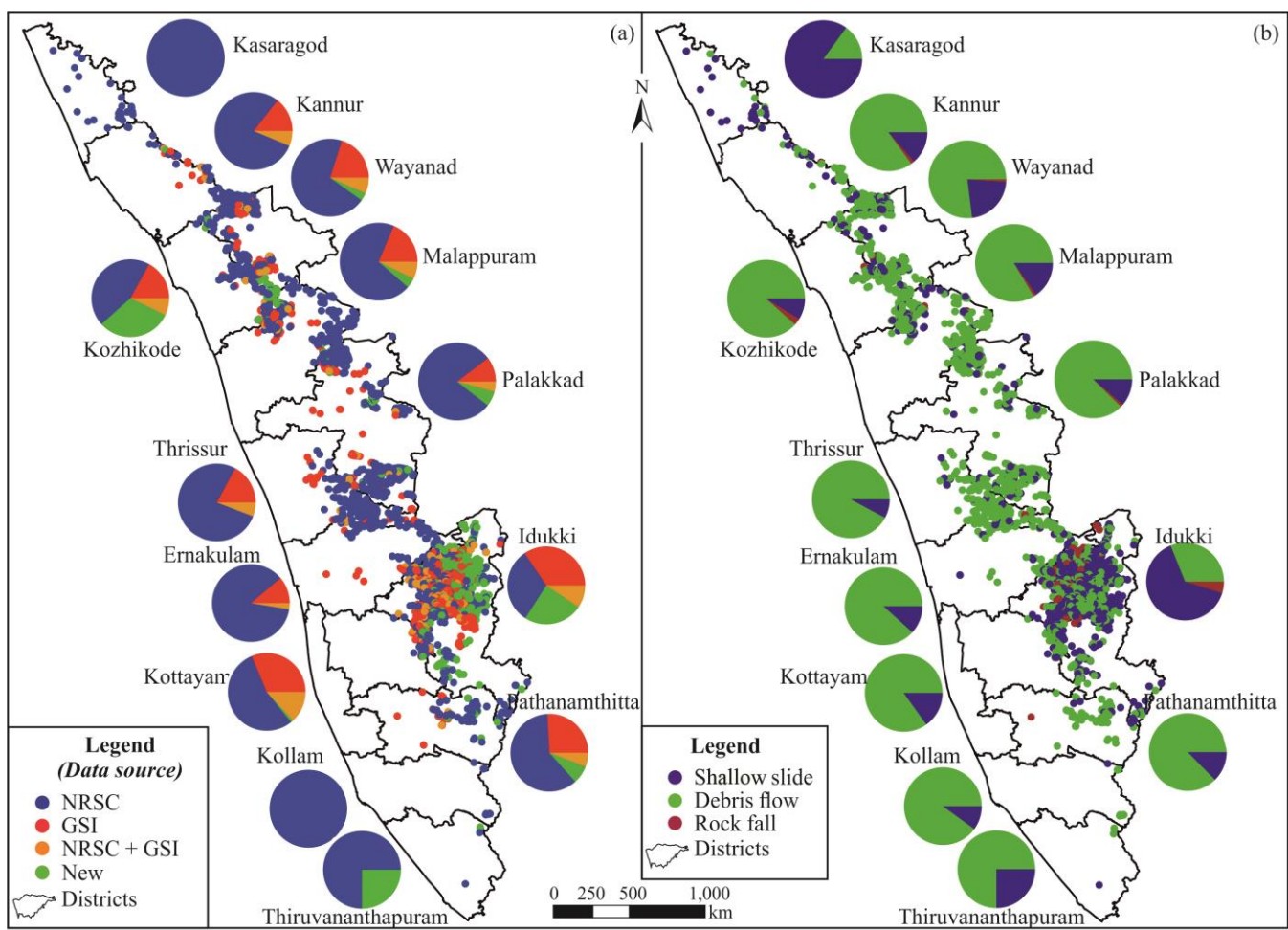

**Figure 15: Map of the final landslide inventory dataset. (a) distribution according to the source of the data; (b) distribution of different landslide types.**

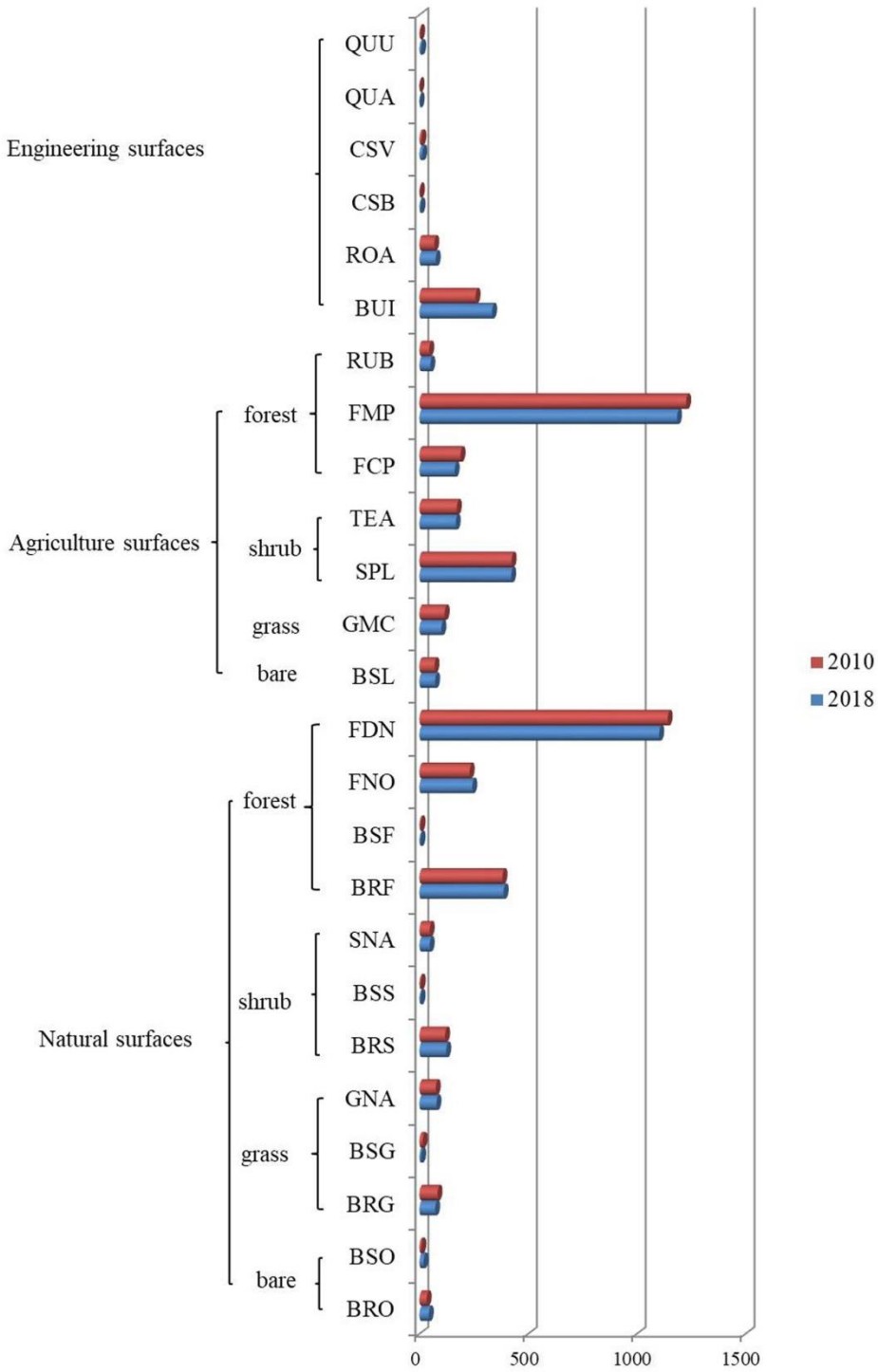

**Figure 16: General land use types of all landslides in Kerala (QUU-quarry in use, QUA-quarry abandoned, CSV-vegetated cut slopes, CSB-bare cut slopes, ROA-roads, BUI-buildings, RUB-rubber plantation, FMP- mixed forest plantation, FCP-forest plantation, TEA-tea plantation, SPL-shrub plantation, GMC-meadows (refers to cultivated grassland), BSL-bare farmland, FDN-dense natural forest, FNO-open natural forest, BSF- bare soil with isolated forests, BRF-bare rock with isolated forests, SNA-natural shrub land, BSS-bare soil with isolated shrubs, BRS-bare rock with isolated shrubs, GNA-natural grass land, BSG-bare soil with isolated grass, BRG-bare rock with isolated grass, BSO-bare soil, BRO-bare rock)**

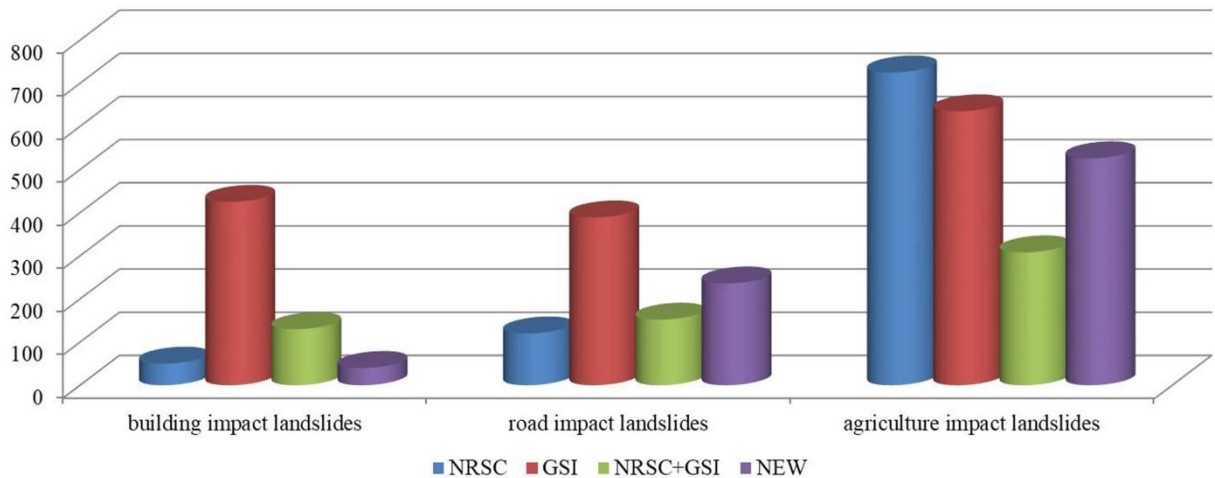

**Figure 17: Damaging landslides of different source.**

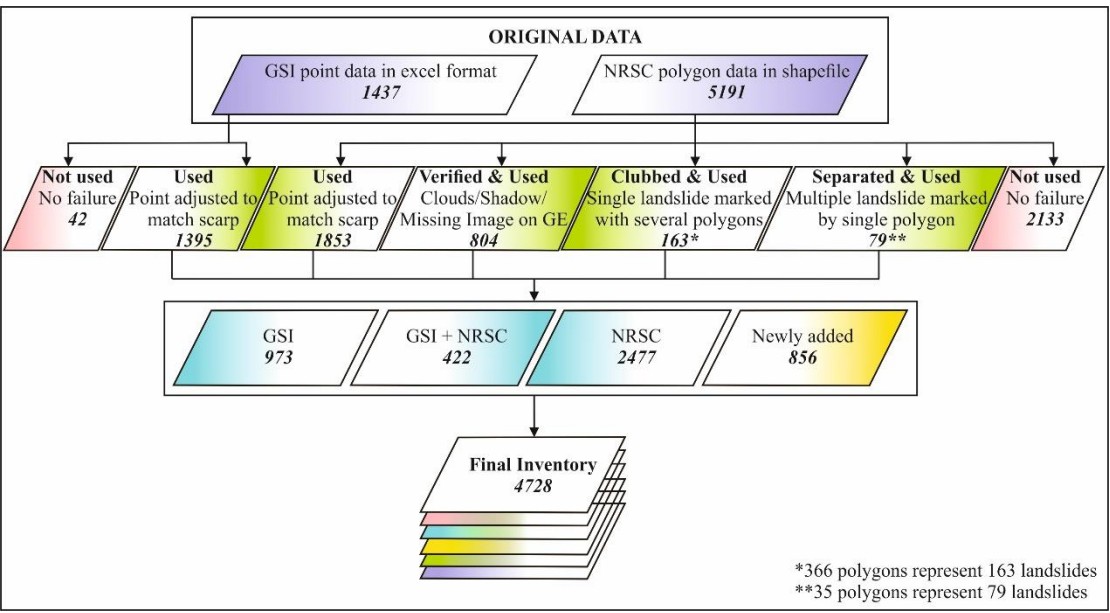

**Figure 18: Overview of the procedure to generate the complete landslide inventory with the number of landslides indicated.**

