# Peer review of "Constructing a complete landslide inventory dataset for the 2018 Monsoon disaster in Kerala, India, for land use change analysis"

_Earth System Science Data, 2020_

## Referee Comment (RC1) · Anonymous Referee #1 · 1 Aug 2020

This is a well-written paper about different scales landslides occurred after triggering events during monsoon in the SW region of India. The material and the methodology are adequate and well explored. The final result is a very interesting and comprehensive database that can be very useful in planning and management of land use in inland Kerala region. The introduction explores comprehensively the available literature although the objectives in mind are not very clear. The figures are very didactic and of great quality. In the section 3.1 Landslide mapping, the authors should consider to clarify the following points: Please, indicate what LULC means (land use land cover?). Despite the difficulty in identifying landslide from photos and sometimes establish elementary diagnostic features, it will be very helpful if the authors explain how superficial

slide (SS) can be distinguished from debris flow (DF); and the latter from rock fall (RF), considering the different sources at stake. Bearing in mind that the images have too coarse resolution for a proper correlation with mass movements, how you can discriminate the different landslides? Could the authors give comprehensive explanations why the majority of the landslides, in the 2018 monsoon event, occurred mainly within forest areas? This behavior is mentioned several times, included in the 6 Discussion and conclusions Taking this aspect into account, the authors might include previously, in the section 2.1 Study area, a description regarding the topographic and orographic characteristics of the entire zone. As you know, the slope is a critical issue for this type of movements in addition to a major triggering event. At a same time, those areas are not attractive for human activities. Please, check bibliographic list. Lines 314 and 316: the references Sahana (2019); Ramachandra and Bharat (2019) are not listed.

---

## Referee Comment (RC2) · Anonymous Referee #2 · 5 Aug 2020

The proposed manuscript intends to provide a complete landslide inventory for the 2018 Moonsoon disaster in Kerala, India, by merging two already available landslide inventories, improving upon it by verifying and correcting the available information using GoogleEarth images and satellite data. Landslide inventories are essential for susceptibility and risk mapping, therefore the presented dataset has the potential to be significant and useful. The overall structure of the article is repetitive and sometimes difficult to follow. Although authors repeat information, they don't present it in a clear way, raising questions to the reader that are answered ahead. One example is the methods section, where only at the end it is clear how the final inventory was able to accommodate both initial datasets and how errors were identified and field data helped in deciding if it was landslide or not. Methodology is presented initially in terms of the encountered challenges, but instead should be presented as a final methodology that can deal with cataloguing challenges. This needs to be improved. English is sound but sometimes difficult to follow as is too colloquial for a research article. State of the art is incomplete in terms of literature about the specific types of landslides (shallow/debris flows) and trigger event (rainfall). Also, alongside with the inventory mapping objective, authors try to provide a susceptibility assessment of the landslides without stating it clearly and without a sound methodology. They provide a section - land use attributes, which has no dataset available and comment in intersecting both datasets (landslides inventory and land use). This is also confusing when dealing with a MS about landslide inventorying. This analysis belongs to a dedicated susceptibility assessment paper. The methodology itself is not new, but although primarily based on the merge of already available catalogues, it completes and verifies the first information. But only the use of the Google images is explained, all about the satellite images is disregard and only mentioned superficially. Also, no additional information on dates and type of images (e.g. resolution) both from Google Earth and satellite is provided. This is important for the discussion and the challenges encountered. About the inventory itself, authors propose to catalog only the initial fail called landslide scarp. But this is very difficult to assess in shallow landslides. The landslides types occurred in Kerala seem to belong to the type shallow landslides/debris flows, which is common in rainfall triggered landslides, but authors to not comment on how this may affect the proposed methodology. Also, the inventory consists in only points, but polygons were generated to give the information on area, for example. The polygons could have been provided as well as part of the dataset, enriching it. I would suggest authors to look for specific literature on rainfall landslide inventories to answer for these questions, particularly in soil-slips and shallow landslides. This is also important for the discussion were authors claim small landslides could not be mapped.

Overall advice is to reformulate the manuscript keeping the methodology simple and clear and only about the inventory mapping, discussing errors and uncertainties inline with literature review.

Dataset: The dataset is new, because it merges and verifies 2 already available land-slide inventories, although it is not clear if the original datasets are freely available. Shapefile metadata should be filled up. What are files Kerela landslides.mid and .mif? Could not open them. A document stating what the different files are should be provided.

Technical corrections: Figure 6 – there are no landslides and markers are the same in both images Figure 8 – There is no landslide. Landslide scar is located right an up from both the idenfied polygon and pinpoint. Figure 9 – One can not infer anything from the images – field work? Caption needs to be improved.

Please also note the supplement to this comment:
https://essd.copernicus.org/preprints/essd-2020-83/essd-2020-83-RC2-supplement.pdf

**Supplement:**

[revised manuscript text omitted]

---

## Author Comment (AC1) · 21 Sep 2020

Thanks for all the constructive comments and suggestions. We have revised the manuscript and responded to all the comments. Details of the responses were in the supplement document.

Comment 1: "The introduction explores comprehensively the available literature although the objectives in mind are not very clear."

Response: Thanks for your constructive comments. Our main objective as stated in the last sentences of the introduction: "The main objective of the study is to develop

a comprehensive event-based landslide inventory database for the 2018 Monsoon in Kerala, that can be used to analyze to what extent these landslides were affected by land use changes." In order to make the introduction section more clearly, some redundant description were deleted, while others related to our methodology description were added. Details of the revisions are as following:

1) The original Line 55 to Line 59 of "such as supervised classification (Lacroix et al., 2013), Object Based Image Analysis (OBIA)(Behling et al., 2014;Casagli et al., 2016; Keyport et al., 2018; Lahousse et al., 2011; Mohan Vamsee et al., 2018), Markov random fields (Lu et al., 2019; Qin et al., 2018), random forests (Stumpf and Kerle, 2011), support vector and other machine learning methods (Lei et al., 2019) or a combination of various algorithms (Aksoy and Ercanoglu, 2012; Li et al., 2016; Lu et al., 2011; Stumpf and Kerle, 2011)." was deleted. The related references were also deleted.

2) Other irrelevant contents such as "UAV, SAR, InSAR" in the original line 59 and 60 were also deleted.

3) "The availability of multi-sourced and multi-temporal high resolution satellite images (HRSI) on the Google Earth platform with 3D viewing capabilities (Crosby, 2012; Fisher et al., 2012) offered major advantages for landslide inventory mapping (Mohammadi et al., 2018). Many authors have generated landslide inventories using the Google Earth platform (Rabby and Li, 2019; Sato and Harp, 2009; Fiorucci et al., 2011; Borrelli et al., 2015). It has also proven to be possible to map event-based landslides by comparing images before and after the event using Google Earth history Viewer (Xu et al., 2014a, 2014b). However, recognizing and mapping specific types of landslides such as rainfall triggered shallow landslides over large areas can be still challenging when using automated techniques. Field verification is only feasible for a limited number of landslides, as it is time and labor intensive, and many landslides may be difficult to access. Therefore, visual image interpretation using HRSI from different time periods may be the best solution. Landslide mapping and classification requires mapping experience and the availability of high resolution images in three-dimension views, either

using stereo images, or oblique views such as in Google Earth, allowing to recognize the specific diagnostic features (Soeters and van Westen, 1996; Zieher et al., 2016)." was added in the introduction section. Also the related references were added in the reference list.

4) The order of some sentences was also modified in paragraphs in the introduction section.

5) The related references was added in the bibliographic list.

comment 2: "In the section 3.1 Landslide mapping, the authors should consider to clarify the following points: Please, indicate what LULC means (land use land cover?)."

Response: LULC means Land Use and Land Cover in section 3.1 Landslide mapping. The detail of LULC was supplemented in the revisions in the last paragraph of the introduction section as follows: "This manuscript focuses on the generation of the dataset consisting of a detailed landslide inventory with land use/land cover (LULC) information for two periods: shortly before the event, and almost a decade older."

comment 3: "Despite the difficulty in identifying landslide from photos and sometimes establish elementary diagnostic features, it will be very helpful if the authors explain how superficial slide (SS) can be distinguished from debris flow (DF); and the latter from rock fall (RF), considering the different sources at stake."

Response: Thank you for this comment. We have added the following sentences to description how to discrimination of SS, DF and RF. Also the related reference for the discrimination was added in the reference list. "Based on the diagnostic features described in Soeters and van Westen (1996), debris flow (DF) features were differentiated from shallow landslides (SS) by the presence of a runout zone, often reaching to the nearest stream, which is not the case for SS. Rock fall features (RF) can be differentiated from the other two processes as they occur on very steep and bare rocky slopes"

comment 4: "Bearing in mind that the images have too coarse resolution for a proper

correlation with mass movements, how you can discriminate the different landslides?"

Response: Landslides were mapped from multi-temporal sub-meter high-resolution images in Google Earth's history viewer, which allowed us to differentiate the three types of landslides according to their discernible image features. Details of discrimination of the three types of landslides were mentioned in the response to comment 3.

comment 5: "Could the authors give comprehensive explanations why the majority of the landslides, in the 2018 monsoon event, occurred mainly within forest areas? This behavior is mentioned several times, included in the 6 Discussion and conclusions Taking this aspect into account, the authors might include previously, in the section 2.1 Study area, a description regarding the topographic and orographic characteristics of the entire zone. As you know, the slope is a critical issue for this type of movements in addition to a major triggering event. At a same time, those areas are not attractive for human activities."

Response: This paper is mainly focusing on the generation of the comprehensive landslide inventory after the 2018 monsoon event. We are writing a follow-up paper in which we analyzed the land use / land cover changes and evaluate these changes. As this paper is focused on the data collection part, we do not give such a comprehensive explanation in this manuscript. The topographic and orographic characteristics of the study were given in the "2.1 Study area" section. However, we would like to answer to this comment briefly here. There are two reasons that the majority of the landslides occurred mainly within forest areas in the 2018 monsoon event.

1) The land use map of 2010 from KSDMA (Kerala State Disaster Management Agency) used in this research as a reference to distinguish natural forest from forest plantation, indicates that the forest coverage rate (natural forest and forest plantation) is 76.51% in this area. The forest coverage in steep areas is even higher, which are more susceptible to landslides. Also, a large part of Kerala is covered by forest (natural forest or forest plantation) confirmed by literatures (Kumar 2005; Roy et al. 2015; Kale

et al. 2016; Sudhakar Reddy et al. 2016; Fox et al. 2017; Ramachandra and Bharath 2019).

2) Physiographically, the west of Kerala consists of coastal plains covered mainly with built up areas and agriculture such as paddies. The eastern part of Kerala is within the Western Ghats with rugged mountains and deep valleys covered with forest and forest plantation. Most of the landslides triggered in the 2018 monsoon event thus were in the eastern mountainous parts of Kerala.

comment 6: "Please, check bibliographic list. Lines 314 and 316: the references Sahana (2019); Ramachandra and Bharat (2019) are not listed."

Response: Thank you for pointing this out. We checked all the references and added the missing references. The references of Sahana (2019), Ramachandra and Bharat (2019) were supplemented in the bibliographic list.

Please also note the supplement to this comment:
https://essd.copernicus.org/preprints/essd-2020-83/essd-2020-83-AC1-supplement.pdf

**Supplement:**

**Response to the comments from anonymous referee 1:**

Thanks for all the constructive comments and suggestions. We have revised the manuscript and responded to all the comments. These are given in this document. Texts in blue font are comments from referee while texts in red font are revisions in the updated manuscript.

**(1) "The introduction explores comprehensively the available literature although the objectives in mind are not very clear."**

Thanks for your constructive comments. Our main objective as stated in the last sentences of the introduction:

"The main objective of the study is to develop a comprehensive event-based landslide inventory database for the 2018 Monsoon in Kerala, that can be used to analyze to what extent these landslides were affected by land use changes."

In order to make the introduction section more clearly, some redundant description were deleted, while others related to our methodology description were added. Details of the revisions are as following:

➢ The original Line 55 to Line 59 of "such as supervised classification (Lacroix et al., 2013), Object Based Image Analysis (OBIA)(Behling et al., 2014;Casagli et al., 2016; Keyport et al., 2018; Lahousse et al., 2011; Mohan Vamsee et al., 2018), Markov random fields (Lu et al., 2019; Qin et al., 2018), random forests (Stumpf and Kerle, 2011), support vector and other machine learning methods (Lei et al., 2019) or a combination of various algorithms (Aksoy and Ercanoglu, 2012; Li et al., 2016; Lu et al., 2011; Stumpf and Kerle, 2011)." was deleted. The related references were also deleted.

➢ Other irrelevant contents such as "UAV, SAR, InSAR" in the original line 59 and 60 were also deleted.

➢ "The availability of multi-sourced and multi-temporal high resolution satellite images (HRSI) on the Google Earth platform with 3D viewing capabilities (Crosby, 2012; Fisher et al., 2012) offered major advantages for landslide inventory mapping (Mohammadi et al., 2018). Many authors have generated landslide inventories using the Google Earth platform (Rabby and Li, 2019; Sato and Harp, 2009; Fiorucci et al., 2011; Borrelli et al., 2015). It has also proven to be possible to map event-based landslides by comparing images before and after the event using Google Earth history Viewer (Xu et al., 2014a, 2014b). However, recognizing and mapping specific types of landslides such as rainfall triggered shallow landslides over large areas can be still challenging when using automated techniques. Field verification is only feasible for a limited number of landslides, as it is time and labor intensive, and many landslides may be difficult to access. Therefore, visual image interpretation using HRSI from different time periods may be the best solution. Landslide mapping and classification requires mapping experience and the availability of high resolution images in three-dimension views, either using stereo images, or oblique views such as in Google Earth, allowing to recognize the specific diagnostic features (Soeters and van Westen, 1996; Zieher et al., 2016)." was added in the introduction section. Also the related references were added in the reference list.

➢ The order of some sentences was also modified in paragraphs in the introduction section.

➢ The related references was added in the bibliographic list

Borrelli, L., Cofone, G., Coscarelli, R. and Gullà, G.: Shallow landslides triggered by consecutive rainfall events at Catanzaro strait (Calabria–Southern Italy), J. Maps, 11(5), 730–744,

doi:10.1080/17445647.2014.943814, 2015.

Fisher, G. B., Amos, C. B., Bookhagen, B., Burbank, D. W. and Godard, V.: Channel widths, landslides, faults, and beyond: The new world order of high-spatial resolution Google Earth imagery in the study of earth surface processes, Spec. Pap. Geol. Soc. Am., 492, 1–22, doi:10.1130/2012.2492(01), 2012

Jacobson, A., Dhanota, J., Godfrey, J., Jacobson, H., Rossman, Z., Stanish, A., Walker, H. and Riggio, J.: A novel approach to mapping land conversion using Google Earth with an application to East Africa, Environ. Model. Softw., 72, 1–9, doi:10.1016/j.envsoft.2015.06.011, 2015.

Rabby, Y. W. and Li, Y.: An integrated approach to map landslides in Chittagong Hilly Areas, Bangladesh, using Google Earth and field mapping, Landslides, 16(3), 633–645, doi:10.1007/s10346-018-1107-9, 2019.

Soeters, R., van Westen, C. J.: Slope instability recognition, analysis, and zonation. In: A. K. Turner, R. L. Schuster (Eds.), Landslides, investigation and mitigation, Special Report, 247, Transportation Research Board, National Research Council, National Academy Press, Washington D.C., USA, 129-177, 1996.

Zieher, T., Perzl, F., Rössel, M., Rutzinger, M., Meißl, G., Markart, G. and Geitner, C.: A multi-annual landslide inventory for the assessment of shallow landslide susceptibility - Two test cases in Vorarlberg, Austria, Geomorphology, 259, 40–54, doi:10.1016/j.geomorph.2016.02.008, 2016.

**(2) "In the section 3.1 Landslide mapping, the authors should consider to clarify the following points: Please, indicate what LULC means (land use land cover?)."**

LULC means Land Use and Land Cover in section 3.1 Landslide mapping. The detail of LULC was supplemented in the revisions in the last paragraph of the introduction section as follows:

➢ "This manuscript focuses on the generation of the dataset consisting of a detailed landslide inventory with land use/land cover (LULC) information for two periods: shortly before the event, and almost a decade older."

**(3) "Despite the difficulty in identifying landslide from photos and sometimes establish elementary diagnostic features, it will be very helpful if the authors explain how superficial slide (SS) can be distinguished from debris flow (DF); and the latter from rock fall (RF), considering the different sources at stake."**

Thank you for this comment. We have added the following sentences to description how to discrimination of SS, DF and RF. Also the related reference for the discrimination was added in the reference list.

➢ "Based on the diagnostic features described in Soeters and van Westen (1996), debris flow (DF) features were differentiated from shallow landslides (SS) by the presence of a runout zone, often reaching to the nearest stream, which is not the case for SS. Rock fall features (RF) can be differentiated from the other two processes as they occur on very steep and bare rocky slopes"

➢ "Soeters, R., van Westen, C. J.: Slope instability recognition, analysis, and zonation. In: A. K. Turner, R. L. Schuster (Eds.), Landslides, investigation and mitigation, Special Report, 247, Transportation Research Board, National Research Council, National Academy Press, Washington D.C., USA, 129-177, 1996."

**(4) "Bearing in mind that the images have too coarse resolution for a proper correlation with mass movements, how you can discriminate the different landslides?"**

Landslides were mapped from multi-temporal sub-meter high-resolution images in Google Earth's history viewer, which allowed us to differentiate the three types of landslides according to their discernible image features. Details of discrimination of the three types of landslides were mentioned in response (3).

**(5) "Could the authors give comprehensive explanations why the majority of the landslides, in the 2018 monsoon event, occurred mainly within forest areas? This behavior is mentioned several times, included in the 6 Discussion and conclusions Taking this aspect into account, the authors might include previously, in the section 2.1 Study area, a description regarding the topographic and orographic characteristics of the entire zone. As you know, the slope is a critical issue for this type of movements in addition to a major triggering event. At a same time, those areas are not attractive for human activities."**

This paper is mainly focusing on the generation of the comprehensive landslide inventory after the 2018 monsoon event. We are writing a follow-up paper in which we analyzed the land use / land cover changes and evaluate these changes. As this paper is focused on the data collection part, we do not give such a comprehensive explanation in this manuscript. The topographic and orographic characteristics of the study were given in the "2.1 Study area" section.

However, we would like to answer to this comment briefly here. There are two reasons that the majority of the landslides occurred mainly within forest areas in the 2018 monsoon event.

➢ The land use map of 2010 from KSDMA (Kerala State Disaster Management Agency) used in this research as a reference to distinguish natural forest from forest plantation, indicates that the forest coverage rate (natural forest and forest plantation) is 76.51% in this area. The forest coverage in steep areas is even higher, which are more susceptible to landslides. Also, a large part of Kerala is covered by forest (natural forest or forest plantation) confirmed by literatures (Kumar 2005; Roy et al. 2015; Kale et al. 2016; Sudhakar Reddy et al. 2016; Fox et al. 2017; Ramachandra and Bharath 2019).

➢ Physiographically, the west of Kerala consists of coastal plains covered mainly with built up areas and agriculture such as paddies. The eastern part of Kerala is within the Western Ghats with rugged mountains and deep valleys covered with forest and forest plantation. Most of the landslides triggered in the 2018 monsoon event thus were in the eastern mountainous parts of Kerala.

**(6) "Please, check bibliographic list. Lines 314 and 316: the references Sahana (2019); Ramachandra and Bharat (2019) are not listed."**

Thank you for pointing this out. We checked all the references and added the missing references. The references of Sahana (2019), Ramachandra and Bharath (2019) were supplemented in the bibliographic list as following:

"Ramachandra, T. V. and Bharath, S.: Carbon Sequestration Potential of the Forest Ecosystems in the Western Ghats, a Global Biodiversity Hotspot, Nat. Resour. Res., doi:10.1007/s11053-019-09588-0, 2019."

"Sahana G.: Kerala floods: Unpacking the reasons for heavy, sustained rainfall, MONGABAY, https://india.mongabay.com/2019/08/kerala-floods-unpacking-the-reasons-for-heavy-sustaine

d-rainfall/, 2019."

---

## Author Comment (AC2) · 21 Sep 2020

Thanks for all the constructive comments and suggestions. We have revised the manuscript and responded to all the comments. Details of the responses were in the supplement document.

Comment 1: "Methodology is presented initially in terms of the encountered challenges, but instead should be presented as a final methodology that can deal with cataloguing challenges. This needs to be improved."

Response: Thank you for your constructive comments. We have now removed the de-

scription of the problems that we encountered when comparing the existing inventories with high resolution images from the Methodology section and created a new section: 2.3 Problems with the use of existing inventories. Also, we reformulated the original methodology section so that it focuses on the work done more clearly. The revised structure of "3 Methodology" is as following:

3 Methodology

3.1 Workflow

3.2 Landslide mapping

3.3 Land use attributes

Comment 2: "English is sound but sometimes difficult to follow as is too colloquial for a research article."

Response: We have tried to improve this and modified section that seemed too colloquial.

Comment 3: "State of the art is incomplete in terms of literature about the specific types of landslides (shallow/debris flows) and trigger event (rainfall)."

Response: We have added more text in the introduction on the mapping of landslide types and issues related to the trigger event. Besides, in order to make the introduction section more clearly and focusing on the methodology, some redundant description was deleted. Details of the revisions are as following:

1) The original Line 55 to Line 59 of "such as supervised classification (Lacroix et al., 2013), Object Based Image Analysis (OBIA)(Behling et al., 2014;Casagli et al., 2016; Keyport et al., 2018; Lahousse et al., 2011; Mohan Vamsee et al., 2018), Markov random fields (Lu et al., 2019; Qin et al., 2018), random forests (Stumpf and Kerle, 2011), support vector and other machine learning methods (Lei et al., 2019) or a combination of various algorithms (Aksoy and Ercanoglu, 2012; Li et al., 2016; Lu et al., 2011;

Stumpf and Kerle, 2011)." was deleted. The related references were also deleted.

2) Other irrelevant contents such as "UAV, SAR, InSAR" in the original line 59 and 60 were also deleted.

3) "The availability of multi-sourced and multi-temporal high resolution satellite images (HRSI) on the Google Earth platform with 3D viewing capabilities (Crosby, 2012; Fisher et al., 2012) offered major advantages for landslide inventory mapping (Mohammadi et al., 2018). Many authors have generated landslide inventories using the Google Earth platform (Rabby and Li, 2019; Sato and Harp, 2009; Fiorucci et al., 2011; Borrelli et al., 2015). It has also proven to be possible to map event-based landslides by comparing images before and after the event using Google Earth history Viewer (Xu et al., 2014a, 2014b). However, recognizing and mapping specific types of landslides such as rainfall triggered shallow landslides over large areas can be still challenging when using automated techniques. Field verification is only feasible for a limited number of landslides, as it is time and labor intensive, and many landslides may be difficult to access. Therefore, visual image interpretation using HRSI from different time periods may be the best solution. Landslide mapping and classification requires mapping experience and the availability of high resolution images in three-dimension views, either using stereo images, or oblique views such as in Google Earth, allowing to recognize the specific diagnostic features (Soeters and van Westen, 1996; Zieher et al., 2016)." was added in the introduction section. Also the related references were added in the reference list.

4) The order of some sentences was also modified in paragraphs in the introduction section.

5) Related references were added in the bibliographic list.

Comment 4: "Also, alongside with the inventory mapping objective, authors try to provide a susceptibility assessment of the landslides without stating it clearly and without a sound methodology"
Response: Our paper does not intend to provide a landslide susceptibility assessment. This is not stated in the objectives, and therefore we also do not provide a methodology for it, nor do we present a landslide susceptibility map. We only aim to develop a dataset from which it will be possible to analyze the effect of land use/land cover change on the landslides that occurred in 2018.

Comment 5: "They provide a section - land use attributes, which has no dataset available and comment in intersecting both datasets (landslides inventory and land use). This is also confusing when dealing with a MS about landslide inventorying. This analysis belongs to a dedicated susceptibility assessment paper."

Response: The section on land use attributes is part of the methodology as indicated in Figure 13 and described in the original section 3.2. Not only were the landslide locations determined based on the comparison of the pre-and post-event Google Earth Images, but also the land use was interpreted and recorded as attributes for the landslide point in 2010 and 2018. The descriptions of the land use/land cover in 2018 (before the monsoon) and in 2010 are part of the methodology, and the land use/land cover attributes are part of the landslide inventory dataset. They were provided as attributes of landslide dataset instead of separate land use/land cover dataset because of the following reasons.

1) The ultimate aim of the study is to analyse to what extent the 2018 landslides were affected by land use changes. To study this, we needed the exact land use/ land cover situation in the initiation areas. As explained in section 3.3, the available land use/land cover products for the Kerala are of insufficient detail and accuracy to use in combination with the mapped landslide points.

2) As is shown in Fig 1 (details of Fig 2 can be seen clearly in the supplement document), the attributes of LU_2010 (land use in 2010 of a landslide initiated area) and LU_2018 (land use in 2018 of a landslide initiated area) were land use information in the manuscript loaded as attributes in the final dataset, which can be used to analyze
the relation between landslide and land use.

3) We have added a sentence in the methodology section 3.1 to clarify that further: "For each landslide we visually interpreted the LULC types using the Google Earth history viewer, for two time periods: before the monsoon of 2018, and for the oldest and nearly complete cover of HRSI for Kerala, which dates back to 2010. Our final landslide inventory dataset was made as points, which were carefully located on the initiation point of the landslides, with attributes related to the landslide type, and the LULC in 2010 and 2018. Due to large number of landslides in the inventory it was not possible to map the landslides as polygons, separating initiation, runout and accumulation areas (Soeters and van Westen, 1996)."

4) The actual analysis of the results is done in another paper which focuses indeed on the evaluation of the land use / land cover changes, and explanation of the causal factors for the landslides. This paper is under preparation and will be submitted to Catena.

Comment 6: "The methodology itself is not new, but although primarily based on the merge of already available catalogues, it completes and verifies the first information. But only the use of the Google images is explained, all about the satellite images is disregard and only mentioned superficially"

Response: Thanks for your comments. The Google Earth Images were our main source of information. As the study area is very large (covering the entire state of Kerala), and is affected by frequent cloud coverage, we would need to acquire a large number of very high resolution satellite images to carry out the study, especially because we did the comparison also for the land use/land cover in the past. As Google Earth provides such high resolution images free of charge, it was used as the main data source. Other satellite data (Resourcesat-2 LISS-IV images, with a spatial resolution of 5.8 m from NRSC) were used for those locations where post-event satellite images in Google Earth were distorted, obscured or missing. We revised the texts in

section 3.1 and the original figure 9, 11, and 12 to illustrate the other satellite images used in this study.

Also, we modified the original figure 9 (Figure 7 in the revised manuscript), figure 11(Figure 9 in the revised manuscript) and figure 12 (Figure 10 in the revised manuscript). Adding Resourcesat-2 LISS-IV images when the high resolution images post the event on Google Earth images were missing.Details of the updated figures and captions can be seen in th supplement document.

Comment 7: "Also, no additional information on dates and type of images (e.g. resolution) both from Google Earth and satellite is provided. This is important for the discussion and the challenges encountered"

Response: The Google Earth images available for the state of Kerala were of varying dates, but we selected those closest to the monsoon event (pre-and-post) and those that were from 2010. The actual dates were different in each part of the state, and cannot be indicated separately. This is also the case for the ResourceSat-2 LISS—IV images, which were used for the areas where Google Earth images were not available. For the resolution of the multi-sourced images, we have indicated this in the following text.

1) "After combining the above-mentioned inventories and overlaying them on multi-sourced sub-meter satellite images for both the pre-and post the event in Google Earth platform (Jacobson et al., 2015; Rabby and Li, 2019), several problems with the data were discovered through visual interpretation."

2) The original Line 133: "as the NRSC data was mainly based on Resourcesat-2 LISS IV images with 5.8 m spatial resolution"

3) "we decided to correct and edit all landslides using visual interpretation based on multi-temporal HRSI available before and after the event on the Google Earth platform. These images with varying dates allow recognizing details in landforms, and land use.

For those areas where the post-event images in Google Earth were distorted, obscured or missing, we used Indian Resourcesat-2 LISS-IV images (with a spatial resolution of 5.8 m and three bands of green, red and near infrared) for the earliest available post-monsoon period of 2018, which were obtained from the NRSC."

4) More information about satellite images on Google Earth in literatures (Jacobson et al., 2015; Rabby and Li, 2019), which were also cited in the manuscript and added in the reference.

Comment 8: "About the inventory itself, authors propose to catalog only the initial fail called landslide scarp. But this is very difficult to assess in shallow landslides. The landslides types occurred in Kerala seem to belong to the type shallow land-slides/debris flows, which is common in rainfall triggered landslides, but authors to not comment on how this may affect the proposed methodology"

Response: We have added sentences in the revised section 3.2 Landslide mapping to explain how we differentiated between the three landslide types:

"The landslides were classified into three simple groups: shallow slide (SS), debris flows (DF) and rock fall (RF). Based on the diagnostic features described in Soeters and van Westen (1996) debris flow (DF) features were differentiated from shallow land-slides (SS) by the presence of a runout zone, often reaching to the nearest stream, which is not the case for SS. Rock fall features (RF) can be differentiated from the other two processes as they occur on very steep and bare rocky slopes."

Comment 9: "Also, the inventory consists in only points, but polygons were generated to give the information on area, for example. The polygons could have been provided as well as part of the dataset, enriching it. I would suggest authors to look for specific literature on rainfall landslide inventories to answer for these questions, particularly in soil-slips and shallow landslides. This is also important for the discussion were authors claim small landslides could not be mapped."

Response: The inventory from NRSC, which was generated through automated image classification, contained polygons. But as we indicated in the paper, these contained too many errors in order to be used in a subsequent analysis. Our visual analysis of the landslide initiation areas was done using points only, because mapping the landslides as polygons would have been too time consuming considering the large study area.

Also for the purpose of our study: the analysis of the effect of land use changes on the occurrence of landslides in the monsoon of 2018, the mapping of points in the initiation areas was adequate. Landslide areas were obtained by measuring the width and length of the landslide. Small landslides that could not be recognized on the very high resolution images, could not be mapped.

Comment 10: "Overall advice is to reformulate the manuscript keeping the methodology simple and clear and only about the inventory mapping, discussing errors and uncertainties inline with literature review."

Response: The ultimate aim of our study is to use the inventory for the analysis of land use/land cover changes, and the description of this is an important component of this paper. The mapping of the land use in the landslide locations for two time periods is an essential part of this paper, and is a more accurate approach than overlaying the landslide inventory on two land use maps from two different periods.

We have reformulated the methodology section to keep it clear on inventory mapping. Details were shown in response to comment 1.

The errors and uncertainties were supplemented in the last paragraph of section 4.2. Also, the final paragraph of the original manuscript is on the uncertainties and completeness. Revisions added are as following. Table 3 was shown in the supplement document and the revised manuscript.

"In the final landslide point dataset, 1276 (27%) out of 4728 landslides were confirmed only by one source, while a total of 3452 (73%) landslides were confirmed by at least
two independent sources (Table 3). Among the single sourced 1276 landslides, 420 (9%) landslides without an estimation of the area of the landslides, as those were the points from GSI for which no area could be determined in the images, because the landslides were too small. These 420 landslides were mapped by GSI as they caused damage to buildings and roads, but could not be identified on Google Earth or Resourcesat-2 satellite images, due to the small size or sheltering by buildings, trees, and clouds. Still, they are accepted in the final dataset because they were visited by geologists in the field. The rest of 856 (18%) single sourced landslides were identified and confirmed by their clear signs on multi-temporal Google Earth images, and about 25 of these were confirmed by field investigation by the authors in May, 2019. Therefore the minimum overall accuracy of the final inventory is 73%, although we consider it to be much larger, given the fact that we visually inspected the entire area. However, it is not possible to quantify the completeness of the final inventory, due to the lack of another independent and confirmed complete inventory. " Details of Table 3 were shown in the supplement docutment.

Comment 11: "Dataset: The dataset is new, because it merges and verifies 2 already available landslide inventories, although it is not clear if the original datasets are freely available."

Response: The NRSC dataset can be consulted but not downloaded. We added a line to the first paragraph of section 2.2: "The landslide dataset can be consulted on the Bhuvan web-platform of NRSC (https://bhuvan-app1.nrsc.gov.in/disaster/disaster.php?id=landslide_monitor ). "

The dataset from GSI is not freely available.

Comment 12: "Shapefile metadata should be filled up."

Response: Metadata.pdf file on DANS (https://doi.org/10.17026/dans-x6c-y7x2) (Fig 2) was provided for the description of shapefile. Details of Fig 2 can be seen clearly in the supplement document.

Comment 13: "What are files Kerela landslides.mid and .mif? Could not open them."

Response: The files with extension .MID and .MIF are MapInfo exchange format, and were automatically converted by the data management organization (DANS) as they consider it to be a better standard than Shapefiles.

Using MapInfo, .MID and .MIF files will be opened.

Comment 14: "A document stating what the different files are should be provided."

Response: The file called metadata.pdf on DANS (Fig. 2) gives the full description of the data.

Comment 15: Response to the technical corrections of Figure 6, Figure 8, and Figure 9

Response: The technical corrections of Figure 6, Figure 8, and Figure 9 were revised. Details of these figures and the revised captions were shown in the supplement document and the revised manuscript.

Comment 16: "Please also note the supplement to this comment: https://essd.copernicus.org/preprints/essd-2020-83/essd-2020-83-RC2-supplement.pdf."

Response: Thank you very much for this comment. We have taken all your valuable suggestions into account. Details were shown in the supplement document and the revised manuscript.

Please also note the supplement to this comment:
https://essd.copernicus.org/preprints/essd-2020-83/essd-2020-83-AC2-supplement.pdf

[Figure]

**Fig. 1.** Fig. 1 Land use attributes in the attribute table of landslide dataset (LU_2010 and LU_2018 in cyan are land use type in 2010 and 2018 for each landslide initiate area, respectively

Interactive
comment

[Figure]

**Fig. 2.** Fig. 2 Metadata to the final landslide shapefile dataset in DANS

**Supplement:**

**Responses to the comments from anonymous referee 2:**

Thanks for all the constructive comments and suggestions. We have revised the manuscript and responded to all the comments. These are given in this document. Texts in blue font are comments from referee while texts in red font are revisions in the updated manuscript.

**(1) "Methodology is presented initially in terms of the encountered challenges, but instead should be presented as a final methodology that can deal with cataloguing challenges. This needs to be improved."**

Thank you for your constructive comments. We have now removed the description of the problems that we encountered when comparing the existing inventories with high resolution images from the Methodology section and created a new section: 2.3 Problems with the use of existing inventories. Also, we reformulated the original methodology section so that it focuses on the work done more clearly. The revised structure of "3 Methodology" is as following:

**3 Methodology**
**3.1 Workflow**
**3.2 Landslide mapping**
**3.3 Land use attributes**

**(2) "English is sound but sometimes difficult to follow as is too colloquial for a research article."**

We have tried to improve this and modified section that seemed too colloquial.

**(3) "State of the art is incomplete in terms of literature about the specific types of landslides (shallow/debris flows) and trigger event (rainfall)."**

We have added more text in the introduction on the mapping of landslide types and issues related to the trigger event. Besides, in order to make the introduction section more clearly and focusing on the methodology, some redundant description was deleted. Details of the revisions are as following:

➢ The original Line 55 to Line 59 of "such as supervised classification (Lacroix et al., 2013), Object Based Image Analysis (OBIA)(Behling et al., 2014;Casagli et al., 2016; Keyport et al., 2018; Lahousse et al., 2011; Mohan Vamsee et al., 2018), Markov random fields (Lu et al., 2019; Qin et al., 2018), random forests (Stumpf and Kerle, 2011), support vector and other machine learning methods (Lei et al., 2019) or a combination of various algorithms (Aksoy and Ercanoglu, 2012; Li et al., 2016; Lu et al., 2011; Stumpf and Kerle, 2011)." was deleted. The related references were also deleted.

➢ Other irrelevant contents such as "UAV, SAR, InSAR" in the original line 59 and 60 were also deleted.

➢ "The availability of multi-sourced and multi-temporal high resolution satellite images (HRSI) on the Google Earth platform with 3D viewing capabilities (Crosby, 2012; Fisher et al., 2012) offered major advantages for landslide inventory mapping (Mohammadi et al., 2018). Many authors have generated landslide inventories using the Google Earth platform (Rabby and Li, 2019; Sato and Harp, 2009; Fiorucci et al., 2011; Borrelli et al., 2015). It has also proven to be possible to map event-based landslides by comparing images before and after the event using Google Earth history viewer (Xu et al., 2014a, 2014b). However, recognizing and mapping specific types of landslides such as rainfall triggered shallow landslides over large areas can be still challenging when using automated techniques. Field verification is only feasible for a limited number of landslides, as it is time and labor intensive, and many landslides may be difficult to access. Therefore, visual image interpretation using HRSI from different time periods may be the best solution. Landslide mapping and classification requires

mapping experience and the availability of HRSI in three-dimension views, either using stereo images, or oblique views such as in Google Earth, allowing to recognize the specific diagnostic features (Soeters and van Westen, 1996; Zieher et al., 2016)." was added in the introduction section. Also the related references were added in the reference list.

➢ The order of some sentences was modified in the introduction section.
➢ Related references were added in the bibliographic list.

Borrelli, L., Cofone, G., Coscarelli, R. and Gullà, G.: Shallow landslides triggered by consecutive rainfall events at Catanzaro strait (Calabria–Southern Italy), J. Maps, 11(5), 730–744, doi:10.1080/17445647.2014.943814, 2015.

Fisher, G. B., Amos, C. B., Bookhagen, B., Burbank, D. W. and Godard, V.: Channel widths, landslides, faults, and beyond: The new world order of high-spatial resolution Google Earth imagery in the study of earth surface processes, Spec. Pap. Geol. Soc. Am., 492, 1–22, doi:10.1130/2012.2492(01), 2012

Jacobson, A., Dhanota, J., Godfrey, J., Jacobson, H., Rossman, Z., Stanish, A., Walker, H. and Riggio, J.: A novel approach to mapping land conversion using Google Earth with an application to East Africa, Environ. Model. Softw., 72, 1–9, doi:10.1016/j.envsoft.2015.06.011, 2015.

Rabby, Y. W. and Li, Y.: An integrated approach to map landslides in Chittagong Hilly Areas, Bangladesh, using Google Earth and field mapping, Landslides, 16(3), 633–645, doi:10.1007/s10346-018-1107-9, 2019.

Soeters, R., van Westen, C. J.: Slope instability recognition, analysis, and zonation. In: A. K. Turner, R. L. Schuster (Eds.), Landslides, investigation and mitigation, Special Report, 247, Transportation Research Board, National Research Council, National Academy Press, Washington D.C., USA, 129-177, 1996.

Zieher, T., Perzl, F., Rössel, M., Rutzinger, M., Meißl, G., Markart, G. and Geitner, C.: A multi-annual landslide inventory for the assessment of shallow landslide susceptibility - Two test cases in Vorarlberg, Austria, Geomorphology, 259, 40–54, doi:10.1016/j.geomorph.2016.02.008, 2016.

**(4) "Also, alongside with the inventory mapping objective, authors try to provide a susceptibility assessment of the landslides without stating it clearly and without a sound methodology"**

Our paper does not intend to provide a landslide susceptibility assessment. This is not stated in the objectives, and therefore we also do not provide a methodology for it, nor do we present a landslide susceptibility map. We only aim to develop a dataset from which it will be possible to analyze the effect of land use/land cover change on the landslides that occurred in 2018.

**(5) "They provide a section - land use attributes, which has no dataset available and comment in intersecting both datasets (landslides inventory and land use). This is also confusing when dealing with a MS about landslide inventorying. This analysis belongs to a dedicated susceptibility assessment paper."**

The section on land use attributes is part of the methodology as indicated in Figure 13 and described in the original section 3.2. Not only were the landslide locations determined based on the comparison of the pre-and post-event Google Earth Images, but also the land use was interpreted and recorded as attributes for the landslide point in 2010 and 2018. The descriptions of the land use/land cover in 2018 (before the monsoon) and in 2010 are part of the methodology, and the land use/land cover attributes are part of the landslide inventory dataset. They were provided as attributes of landslide dataset instead of separate land use/land cover dataset because of the following reasons.

- The ultimate aim of the study is to analyse to what extent the 2018 landslides were affected by land use changes. To study this, we needed the exact land use/ land cover situation in the initiation areas. As explained in section 3.3, the available land use/land cover products for the Kerala are of insufficient detail and accuracy to use in combination with the mapped landslide points.
- As is shown in Fig 1, the attributes of LU_2010 (land use in 2010 of a landslide initiated area) and LU_2018 (land use in 2018 of a landslide initiated area) were land use information in the manuscript loaded as attributes in the final dataset, which can be used to analyze the relation between landslide and land use.

| Width | Area | Building_I | Road_impac | Impact_Agr | LU_2010 | LU_2018 | Specific_r | |
|---|---|---|---|---|---|---|---|---|
| 26 | 2548 | 0 | N | FMP | FMP | FMP | | Tv |
| 0 | 647.432 | 0 | N | N | BRF | BRF | | |
| 0 | 3000 | 0 | N | N | BRF | BRF | | |
| 0 | 2962.66 | 0 | N | N | BRF | BRF | | |
| 0 | 5123.88 | 0 | N | N | BRF | BRF | | |
| 0 | 1425.47 | 0 | N | N | BRF | BRF | | |
| 0 | 898.024 | 0 | N | N | BRS | BRS | | |
| 0 | 2447.78 | 0 | N | N | BRS | BRS | | |
| 0 | 42439.9 | 0 | N | N | BRS | BRS | | |
| 0 | 1196.41 | 0 | N | N | BRS | BRS | | |
| 0 | 0 | 1 | N | N | BUI | BUI | | 1 |
| 0 | 3170.33 | 2 | N | N | BUI | BUI | | De |
| 0 | 0 | 0 | C | N | BUI | BUI | | Si |
| 0 | 0 | 1 | N | N | BUI | BUI | | ш |
| 0 | 0 | 1 | N | N | BUI | BUI | | Th |
| 0 | 0 | 0 | N | N | BUI | BUI | | H: |
| 0 | 0 | 1 | N | N | FMP | BUI | 1 | cı |

Fig. 1 Land use attributes in the attribute table of landslide dataset (LU_2010 and LU_2018 in cyan are land use type in 2010 and 2018 for each landslide initiate area, respectively

- We have added a sentence in the methodology section 3.1 to clarify that further:

"For each landslide we visually interpreted the LULC types using the Google Earth history viewer, for two time periods: before the monsoon of 2018, and for the oldest and nearly complete cover of HRSI for Kerala, which dates back to 2010. Our final landslide inventory dataset was made as points, which were carefully located on the initiation point of the landslides, with attributes related to the landslide type, and the LULC in 2010 and 2018. Due to large number of landslides in the inventory it was not possible to map the landslides as polygons, separating initiation, runout and accumulation areas (Soeters and van Westen, 1996)."

- The actual analysis of the results is done in another paper which focuses indeed on the evaluation of the land use / land cover changes, and explanation of the causal factors for the landslides. This paper is under preparation and will be submitted to *Catena*.

**(6) "The methodology itself is not new, but although primarily based on the merge of already available catalogues, it completes and verifies the first information. But only the use of the Google images is explained, all about the satellite images is disregard and only mentioned superficially"**

Thanks for your comments. The Google Earth Images were our main source of information. As the study area is very large (covering the entire state of Kerala), and is affected by frequent cloud coverage, we would need to acquire a large number of very high resolution satellite images to carry out the study, especially because we did the comparison also for the land use/land cover in the past. As Google Earth provides such high resolution images free of charge, it was used as the main data source. Other satellite

data (Resourcesat-2 LISS-IV images, with a spatial resolution of 5.8 m from NRSC) were used for those locations where post-event satellite images in Google Earth were distorted, obscured or missing. We revised the texts in section 3.1 and the original figure 9, 11, and 12 to illustrate the other satellite images used in this study.

➢ We modified the sentence in the revised section of 3.1:

"For those areas where the post-event images in Google Earth were distorted, obscured or missing, we used Indian Resourcesat-2 LISS-IV images (with a spatial resolution of 5.8 m and three bands of green, red and near infrared) for the earliest available post-monsoon period of 2018, which were obtained from the National Remote Sensing Center."

➢ Also, we modified the original figure 9 (Figure 7 in the revised manuscript), figure 11(Figure 9 in the revised manuscript) and figure 12 (Figure 10 in the revised manuscript). Adding Resourcesat-2 LISS-IV images when the high resolution images post the event on Google Earth images were missing. The updated figures with captions are as following:

[Figure]

**Figure 7: Example of a situation where vegetation re-growth made it difficult to identify the scarps on Google Earth images due to the large time span between the event and the first available images within Google Earth. The original NRSC landslide polygons were generated from the classification of Resourcesat-2 LISS-IV images that were taken within 15 days of the event. (a) pre-landslide Google Earth image; (b) earliest available post-landslide Google Earth image, where the landslide cannot be recognized; (c) mapping of the landslide initiation point based on Resourcesat-2 LISS-IV image (RGB combination: near infrared, red, green). Basemap data for a and b © 2019 Google**

[Figure]

**Figure 9: Example of the presence of darks shadows in the post-event images in Google Earth images, making it impossible to check the original NRSC landslide polygons. (a) pre-landslide Google Earth image; (b) earliest available post-landslide Google Earth image, where the landslide cannot be**

**recognized**; **(c) mapping of the landslide initiation point based on Resourcesat-2 LISS-IV image (RGB combination: near infrared, red, green). Basemap data for a and b © 2019 Google**

[Figure]

**Figure 10: Example of the obstruction of view by clouds where the original NRSC landslide polygons could not be checked. (a) pre-landslide Google Earth image; (b) earliest available post-landslide Google Earth image, where the landslide cannot be recognized; (c) mapping of the landslide initiation point based on Resourcesat-2 LISS-IV image (RGB combination: near infrared, red, green). Basemap data for a and b © 2019 Google**

**(7) "Also, no additional information on dates and type of images (e.g. resolution) both from Google Earth and satellite is provided. This is important for the discussion and the challenges encountered"**

The Google Earth images available for the state of Kerala were of varying dates, but we selected those closest to the monsoon event (pre-and-post) and those that were from 2010. The actual dates were different in each part of the state, and cannot be indicated separately. This is also the case for the ResourceSat-2 LISS—IV images, which were used for the areas where Google Earth images were not available. For the resolution of the multi-sourced images, we have indicated this in the following text.

➢ "After combining the above-mentioned inventories and overlaying them on multi-sourced sub-meter satellite images for both the pre-and post the event in Google Earth platform (Jacobson et al., 2015; Rabby and Li, 2019), several problems with the data were discovered through visual interpretation."

➢ The original Line 133: "as the NRSC data was mainly based on Resourcesat-2 LISS IV images with 5.8 m spatial resolution"

➢ "we decided to correct and edit all landslides using visual interpretation based on multi-temporal HRSI available before and after the event on the Google Earth platform. These images with varying dates allow recognizing details in landforms, and land use. For those areas where the post-event images in Google Earth were distorted, obscured or missing, we used Indian Resourcesat-2 LISS-IV images (with a spatial resolution of 5.8 m and three bands of green, red and near infrared) for the earliest available post-monsoon period of 2018, which were obtained from the NRSC."

➢ More information about satellite images on Google Earth in the following literatures, which were also cited in the manuscript and added in the reference list:

Jacobson, A., Dhanota, J., Godfrey, J., Jacobson, H., Rossman, Z., Stanish, A., Walker, H. and Riggio, J.: A novel approach to mapping land conversion using Google Earth with an application to East Africa, Environ. Model. Softw., 72, 1–9, doi:10.1016/j.envsoft.2015.06.011, 2015.

Rabby, Y. W. and Li, Y.: An integrated approach to map landslides in Chittagong Hilly Areas, Bangladesh, using Google Earth and field mapping, Landslides, 16(3), 633–645, doi:10.1007/s10346-018-1107-9, 2019.

**(8) "About the inventory itself, authors propose to catalog only the initial fail called landslide scarp. But this is very difficult to assess in shallow landslides. The landslides types occurred in Kerala seem to belong to the type shallow landslides/debris flows, which is common in rainfall triggered landslides, but authors to not comment on how this may affect the proposed methodology"**

We have added sentences in the revised section 3.2 Landslide mapping to explain how we differentiated between the three landslide types:

"The landslides were classified into three simple groups: shallow slide (SS), debris flows (DF) and rock fall (RF). Based on the diagnostic features described in Soeters and van Westen (1996) debris flow (DF) features were differentiated from shallow landslides (SS) by the presence of a runout zone, often reaching to the nearest stream, which is not the case for SS. Rock fall features (RF) can be differentiated from the other two processes as they occur on very steep and bare rocky slopes."

**(9) "Also, the inventory consists in only points, but polygons were generated to give the information on area, for example. The polygons could have been provided as well as part of the dataset, enriching it. I would suggest authors to look for specific literature on rainfall landslide inventories to answer for these questions, particularly in soil-slips and shallow landslides. This is also important for the discussion were authors claim small landslides could not be mapped."**

The inventory from NRSC, which was generated through automated image classification, contained polygons. But as we indicated in the paper, these contained too many errors in order to be used in a subsequent analysis. Our visual analysis of the landslide initiation areas was done using points only, because mapping the landslides as polygons would have been too time consuming considering the large study area.

Also for the purpose of our study: the analysis of the effect of land use changes on the occurrence of landslides in the monsoon of 2018, the mapping of points in the initiation areas was adequate. Landslide areas were obtained by measuring the width and length of the landslide. Small landslides that could not be recognized on the very high resolution images, could not be mapped.

**(10) "Overall advice is to reformulate the manuscript keeping the methodology simple and clear and only about the inventory mapping, discussing errors and uncertainties inline with literature review."**

➢ The ultimate aim of our study is to use the inventory for the analysis of land use/land cover changes, and the description of this is an important component of this paper. The mapping of the land use in the landslide locations for two time periods is an essential part of this paper, and is a more accurate approach than overlaying the landslide inventory on two land use maps from two different periods.

➢ We have reformulated the methodology section to keep it clear on inventory mapping. Details were shown in response to comment (1).

➢ The errors and uncertainties were supplemented in the last paragraph of section 4.2. Also, the final paragraph of the original manuscript is on the uncertainties and completeness. Revisions added are as following:

"In the final landslide point dataset, 1276 (27%) out of 4728 landslides were confirmed only by one source, while a total of 3452 (73%) landslides were confirmed by at least two independent sources (Table 3). Among the single sourced 1276 landslides, 420 (9%) landslides without an estimation of the area of the landslides, as those were the points from GSI for which no area could be determined in the images, because the landslides were too small. These 420 landslides were mapped by GSI as they caused damage to buildings and roads, but could not be identified on Google Earth or Resourcesat-2 satellite images, due to

the small size or sheltering by buildings, trees, and clouds. Still, they are accepted in the final dataset because they were visited by geologists in the field. The rest of 856 (18%) single sourced landslides were identified and confirmed by their clear signs on multi-temporal Google Earth images, and about 25 of these were confirmed by field investigation by the authors in May, 2019. Therefore the minimum overall accuracy of the final inventory is 73%, although we consider it to be much larger, given the fact that we visually inspected the entire area. However, it is not possible to quantify the completeness of the final inventory, due to the lack of another independent and confirmed complete inventory. "

**Table 3: The number of landslide confirmation by different means in Kerala**

| Confirmation means | GSI only (Field mapping) | Google Earth only (Visual image interpretation) | GSI, Google Earth | NRSC, Google Earth/ Resourcesat-2 LISS-IV | GSI, NRSC, Google Earth |
|---|---|---|---|---|---|
| Number/% | 420/9% | 856/18% | 553/12% | 2477/52% | 422/9% |

**(11) "Dataset: The dataset is new, because it merges and verifies 2 already available landslide inventories, although it is not clear if the original datasets are freely available."**

➢   The NRSC dataset can be consulted but not downloaded. We added a line to the first paragraph of section 2.2:

"The landslide dataset can be consulted on the Bhuvan web-platform of NRSC (https://bhuvan-app1.nrsc.gov.in/disaster/disaster.php?id=landslide_monitor )."

➢   The dataset from GSI is not freely available.

**(12) "Shapefile metadata should be filled up."**

Metadata.pdf file on DANS (https://doi.org/10.17026/dans-x6c-y7x2) (Fig 2) was provided for the description of shapefile.

[Figure]

Fig. 2 Metadata to the final landslide shapefile dataset in DANS

**(13) "What are files Kerela landslides.mid and .mif? Could not open them."**

➢   The files with extension .MID and .MIF are MapInfo exchange format, and were automatically converted by the data management organization (DANS) as they consider it to be a better standard than Shapefiles.

➢ Using MapInfo, .MID and .MIF files will be opened.

**(14) "A document stating what the different files are should be provided."**

The file called metadata.pdf on DANS (Fig. 2) gives the full description of the data.

**(15) Response to the technical corrections of Figure 6, Figure 8, and Figure 9**

➢ **"Figure 6 – there are no landslides and markers are the same in both images"**

This figure contains the landslide boundaries of the original NRSC landslide polygon on the pre- and post-event images, and the interpreted landslide initiation points indicated as markers in the right image. The caption of Figure 6 in the original manuscript was revised to make the description more clearly. Details of the figure and revised caption are as following:

[Figure]

**"Figure 6: Example of a situation where the original NRSC landslide polygon was separated and converted into several landslides, marked by points and digitized on the top of the scarps. (a) pre-landslide image with NRSC landslide polygon on top; (b) post-landslide image, with the NRSC polygon on top, which shows that there are two landslides instead of a single one; (c) the mapping of landslide points in the scarps of the two landslides. Basemap data© 2019 Google "**

➢ **"Figure 8 – there are no landslide. Landslide scar is located right an up from both the idenfied polygon and pinpoint."**

We modified the caption of original Figure 8 (Figure 12 in the revised manuscript) as following:

[Figure]

**"Figure12 (Figure 8 in the original manuscript):** Example of a situation where the original GSI landslide points were shifted to the top of the landslide scarps. (a): pre-event image with landslide points from the GSI inventory; (b): post-event image with original landslide points from the GSI inventory; (c): post-event image with adjusted landslide points. **Basemap data© 2019 Google"**

➢ **"Figure 9 – One can not infer anything from the images – field work? Caption needs to be improved."**

Figure 9 (Figure 7 in the revised manuscript) with its caption in the original manuscript is revised. The basemap of Figure 9c was replaced by Resourcesat-2 LISS IV images. The caption of original Figure 9 was revised to make the description more clearly. The revised Figure 9 (Figure 7 in the revised manuscript) and its caption are as following:

[Figure]

**Figure 7(Figure 9 in the original manuscript):** Example of a situation where vegetation re-growth made it difficult to identify the scarps on Google Earth images due to the large time span between the event and the first available images within Google Earth. The original NRSC landslide polygons were generated from the classification of Resourcesat-2 LISS-IV images that were taken within 15 days of the event. (a) pre-landslide Google Earth image; (b) earliest available post-landslide Google Earth image, where the landslide cannot be recognized; (c) mapping of the landslide initiation point based on Resourcesat-2 LISS-IV image (RGB combination: near infrared, red, green). Basemap data for a and b © 2019 Google

➤ In view of the constructive comments on the above figures, we also modified the caption of the original Figure 7 (Figure 11 in the revised manuscript) to make the description more clearly. The details of this Figure and caption are as following:

[Figure]

**Figure 11(Figure 7 in the original manuscript): Example of a situation where the original GSI landslide points were accepted although there were no manifestation of landslide scarps was visible in pre- and post- event images within Google Earth. We assumed that landslide were properly marked in the field by the surveyors, and that they must have been very small and hidden from view by surrounding vegetation or buildings. Basemap data© 2019 Google**

**(16)** **"Please also note the supplement to this comment: https://essd.copernicus.org/preprints/essd-2020-83/essd-2020-83-RC2-supplement.pdf."**

Thank you very much for this comment. We have taken all your valuable suggestions into account. Examples as:

➤ All "shape file" was replaced as "shapefile".

➤ Also, surficial slide was replaced by "shallow slide" in the whole manuscript.

➤ To avoid repetition and make the description more clearly and concise, high-resolution satellite images was replace by "HRSI" from the second occurrence.

➤ The caption of Table 1 "**The number of landslides per district in Kerala for the various data sources**" was replaced by "**The number of landslides per district in Kerala of the final dataset with various sources**"

---

## Author Response (AR2)

1. There are no words changes in the final uploaded file.

2. Minor font modifications were revised in the uploaded figures. Examples as:

- "a" was revised as "(b)"
- The order of legend and scale of figure 8 were revised.

All revisions were marked in red font.